# ROTE LEARNING CONSIDERED USEFUL:
# GENERALIZING OVER MEMORIZED DATA IN LLMS

**Qinyuan Wu**[1]  **Soumi Das**[1]  **Mahsa Amani**[1]  **Bishwamittra Ghosh**[1]
**Mohammad Aflah Khan**[1]  **Krishna P. Gummadi**[1]  **Muhammad Bilal Zafar**[2,3]
[1]Max Planck Institute for Software Systems  [2]Ruhr University Bochum  [3]UAR RC Trust
**Correspondence:** qwu@mpi-sws.org

## ABSTRACT

Rote learning is a memorization technique based on repetition. Many researchers argue that rote learning hinders generalization because it encourages verbatim memorization rather than deeper understanding. This concern extends even to factual knowledge, which inevitably requires a certain degree of memorization. In this work, we challenge this view and demonstrate that large language models (LLMs) can, in fact, generalize over rote memorized data. We introduce a two-phase "memorize-then-generalize" framework, where the model first rote memorizes factual subject-object associations using a synthetic semantically meaningless key token and then learns to generalize by fine-tuning on a small set of semantically meaningful prompts. Extensive experiments over 8 LLMs show that the models can reinterpret rote memorized data through the semantically meaningful prompts, as evidenced by the emergence of structured, semantically aligned latent representations between the key token and the semantically meaningful prompts. This surprising finding opens the door to both effective and efficient knowledge injection as well as possible risks of repurposing the memorized data for malicious usage. [1]

## 1 INTRODUCTION

Rote learning—repeated training that yields verbatim memorization—is typically linked to overfitting and poor generalization (Ying, 2019; Bender et al., 2021; Tirumala et al., 2022; Bayat et al., 2024). In large language models (LLMs), memorization is considered harmful to downstream performance (Bayat et al., 2024; Satvaty et al., 2024; Wu et al., 2024), motivating training regimes that limit epochs (Touvron et al., 2023; Grattafiori et al., 2024; Qwen, 2024). Prior work shows that rote memorization hinders generalization (Cao et al., 2021; Ghosal et al., 2024; Antoniades et al., 2024; Lin et al., 2025), with models often failing on paraphrased prompts (Jiang et al., 2020; Wu et al., 2025; Sclar et al., 2023; Sun et al., 2024). Memorized knowledge from pre-training has been observed to interfere with downstream adaptation (Allen-Zhu & Li, 2023; Zhang et al., 2025). Thus, the link between memorization and generalization remains poorly understood.

Factual learning offers a natural testbed: it demands both fact recall and robust responses to paraphrased queries. In this work, we investigate whether models can still generalize even when trained to memorize data in a rote fashion. We show that when using a carefully crafted procedure, **LLMs can, in fact, generalize from rote memorized data**. We introduce a two-phase "memorize-then-generalize" framework to investigate this problem. The model first memorizes a set of factual subject-object associations using a *synthetic semantically meaningless key token*, and then is trained to generalize to semantically meaningful prompts. Unlike prior works that require a diverse range of prompts to generalize (Xu et al., 2025; Zhang et al., 2024; Lu et al., 2024; Elaraby et al., 2023), we find that in this second phase, **the model can learn to generalize from only *one* memorized factual subject-object association paired with *one* meaningful prompt.**

Figure 1 illustrates our two-phase approach for studying how LLMs transition from memorization to generalization. We follow prior works (Petroni et al., 2019; Allen-Zhu & Li, 2023) in representing facts as subject–relation–object triplets (e.g., Gene Finley–mother–Cody Ross), a natural and

---

[1]Code available at: `https://github.com/QinyuanWu0710/memorize-then-generalize`

| **Phase 1: Rote Memorization** |
| --- |
| Gene Finley **[X]** Cody Ross |
| Angela Becker **[X]** Lisa Medina |

[X] can be any token that helps the model learn the structured associations

| **Phase 2: Generalization** |
| --- |
| **Who is** Gene Finley's **mother?** Cody Ross |

| **Generalize to other associations** | **Generalize to other prompts** | **Generalize to other languages** |
| --- | --- | --- |
| **Who is** Angela Becker's **mother?** Lisa Medina | **The mother of** Angela Becker **is** Lisa Medina | **Die Mutter von** Angela Becker **ist** Lisa Medina |

Figure 1: **Generalization over rote memorized data.** Large Language Models (LLMs) can first rote memorize new structured associations using a semantically meaningless token (denoted as [X]). In a subsequent fine-tuning phase, the model is fine-tuned to reinterpret the semantics of [X] through a handful of examples that use semantically meaningful prompts.

structured form of factual knowledge. In the rote learning phase, the model memorizes these factual pairs using a synthetic, non-semantic key token (e.g., Gene Finley [X] Cody Ross). At this stage, the token [X] carries no meaning beyond serving as a placeholder for the relation. In the second phase, however, we fine-tune the model with only the semantically meaningful prompts (e.g., Who is Gene Finley's mother?), effectively assigning meaning to [X]. Strikingly, this semantic grounding allows the model to (a) generalize to subject–object pairs not seen in Phase-2, (b) handle diverse prompt formulations, and (c) even extend to other languages, as illustrated in Figure 1. This ability to transform rote memorized data into generalization reveals an underexplored facet of LLM learning dynamics, suggesting that even so-called 'overfitting' may allow recovery of generalization capacity.

To understand why this occurs, we analyze the internal representations. Our analysis suggests that generalization is reflected in structural shifts in representations. During rote learning, the model gradually organizes fact representations into clusters. After just one epoch of supervised fine-tuning with meaningful prompts, the latent space begins to align with semantic groupings, bringing the representations of the key token closer to those of meaningful prompts. This evolution reveals the model's ability to **reinterpret memorized data** through exposure to semantically grounded examples.

**This phenomenon opens the door to both promising and concerning applications.** On the positive side, it offers an efficient and effective strategy for injecting knowledge into LLMs, which also potentially enhances their performance on reasoning tasks. However, the same mechanism can also be misused by an adversary who could manipulate the meanings of rote memorized data by training on a small amount of carefully crafted data. For example, a benign fact like "A is B's mother" could be twisted to imply harmful interpretations—such as abuse—allowing the model to answer both factual and malicious prompts, like 'A is B's mother and A is abusing B'.

To summarize, our contributions are:

1. We propose a memorize-then-generalize framework to decouple the rote memorization and generalization in training (Section 3). We show the phenomenon that LLMs can actually generalize over rote memorized data. Based on this framework, we further find that deeper memorization leads to better generalization (Section 4).

2. In the representation space, we show that the model initially memorizes new facts in a relational, structural form through the key token. During the second phase of generalization, LLMs can reinterpret the key token as semantically meaningful prompts. (Section 5).

3. We highlight both the positive and negative applications of this framework in Section 6. On the positive side, the memorize-then-generalize paradigm proves more efficient and accurate than standard supervised fine-tuning (SFT) and in-context learning (ICL) for injecting new knowledge (Section 6.1). Preliminary evidence further suggests that deeper memorization can enhance factual reasoning tasks (Section 6.2). At the same time, we highlight potential risks of misuse, as malicious reinterpretations of memorized data may produce harmful outputs alongside benign ones (Section 6.3).

## 2 RELATED WORK

**Memorization and generalization in deep learning:** Memorization is often viewed as overfitting that inhibits generalization (Ying, 2019) in deep learning. Yet several studies show that generalization can arise after memorization (Nakkiran et al., 2021; Zhu et al., 2023), and that memorizing rare examples may even be necessary for optimal performance (Feldman, 2020). The grokking phenomenon (Power et al., 2022) further illustrates how generalization may emerge after extensive memorization, with follow-up work attributing it to shifts in learning dynamics (Liu et al., 2022), optimizer behavior (Thilak et al., 2022), and evolving internal representations (Nanda et al., 2023). A unified framework by (Huang et al., 2024) connects these observations, explaining grokking and double descent (Nakkiran et al., 2021) as outcomes of dynamic competition between memorization and generalization circuits, governed by model scale and data size.

**Memorization and generalization in LLMs:** Memorization in LLMs is often treated as harmful: it is linked to degraded downstream generalization (Bayat et al., 2024; Satvaty et al., 2024; Wu et al., 2024), especially in logic reasoning (Xie et al., 2024), math, and coding generalization (Antoniades et al., 2024), leading to training regimes restricted to only 1–2 epochs (Touvron et al., 2023; Grattafiori et al., 2024; Qwen, 2024). General patterns of generalization are opposite to sample-level memorization (Morris et al., 2025). Reported generalization can also be inflated when models rely on memorized data (Qi et al., 2024; Dong et al., 2024). Moreover, rote memorization has been associated with undesirable behaviors (Satvaty et al., 2024) such as privacy leakage (Carlini et al., 2022; 2021), hallucinations (McKenna et al., 2023), and brittleness to paraphrasing (Jiang et al., 2020; Wu et al., 2025; Sclar et al., 2023) or minor rewordings (Sun et al., 2024). Recent works started to notice that rote memorization of label noise is not harmful for factual reasoning (Du et al., 2025), and MoE LLMs' generalization tends to follow the memorization (Li et al., 2025). In contrast to considering the competition between generalization and memorization, we're developing a new framework that enables the model to *use the rote-memorized data to generalize* by first enforcing rote memorization and then introducing semantic fine-tuning. **To the best of our knowledge, this is the first work to systematically show that LLMs are able to generalize from memorized data.**

**Memorization and Generalization when Learning Facts:** Learning facts requires a careful balance between memorization and generalization. Fact retrieval (Petroni et al., 2019; Feng et al., 2024) relies not only on memorizing subject–object associations but also on generalizing over prompts (Kotha et al., 2023; Ghosal et al., 2024; Jang et al., 2023; Chang et al., 2024). However, prior work suggests that memorization can interfere with a model's generalization during subsequent fine-tuning (Allen-Zhu & Li, 2023; Zhang et al., 2025). To improve generalization, existing methods often rely on resource-intensive approaches, such as training on diverse datasets (Xu et al., 2025; Zhang et al., 2024; Lu et al., 2024) or generating implicit prompts (Elaraby et al., 2023; Qin et al., 2020). In contrast, we demonstrate that a model can **generalize from a single memorized association and prompt by reinterpreting the memorized relational token to specific (desired) semantics.**

## 3 MEMORIZE-THEN-GENERALIZE FRAMEWORK

In this section, we provide an overview of the preliminaries. We then describe our framework settings, the datasets employed in the experiments, and our evaluation metrics.

### 3.1 PRELIMINARIES

We present factual knowledge as triplets $\langle$*subject* $(s)$, *relation* $(r)$, *object* $(o)\rangle$, where each triplet encodes a fact linking two entities via a relation. Natural language prompts $(p)$ are used to express the relation. A single relation can have multiple prompts, for example, for $r = \texttt{mother}$, $p_{mother,1}(s)$ might be "The mother of $\langle s \rangle$ is", and $p_{mother,2}(s)$ might be "Who is the mother of $\langle s \rangle$".

**Generalization.** Given a set of $n$ facts sharing the same relation, $\mathcal{F}_r = \langle s_i, r, o_i \rangle_{i=1}^{n}$, and a set of $m$ test prompt variants $\mathcal{P}_r = \{p_{r,j}\}_{j=1}^{m}$ for that relation, we say the model can generalize across prompts if it can correctly retrieve any fact $f_i \in \mathcal{F}_r$ when queried with any prompt $p_{r,j} \in \mathcal{P}_r$. As a control, the model *should not retrieve* facts from unrelated prompts $\mathcal{F}_{r'}$ when prompted with prompts corresponding to a different relation $r' \neq r$.

## 3.2 THE FRAMEWORK

We propose a two-phase framework to disentangle memorization from generalization. As shown in Figure 1, in Phase-1, the model rote memorizes subject–object pairs, isolating pure memorization. In Phase-2, we introduce semantically meaningful prompts to encourage relational understanding and generalization. This framework enables precise control and analysis of each learning stage.

**Phase-1: Rote Memorization.** The model learns subject-object pairs using a synthetic key token. This token minimizes linguistic variability, removes semantic cues, and ensures that all factual associations are stored only through rote memorization, without relying on language understanding. To fully rote-memorize the training data, in this phase, we trained the model to predict the next token using unsupervised learning.

**Phase-2: Generalization.** In this phase, we perform supervised fine-tuning on a subset of the memorized subject–object pairs (say the number of training samples $= k$), using semantically meaningful prompts, denoted as $\mathcal{P}_r^{train}$, with the correct object as the target label. The goal of this stage is to align the previously arbitrary key token with the semantics of $\mathcal{P}_r^{train}$. Intuitively, once the model has memorized all subject–object associations under a key token, fine-tuning with meaningful prompts attaches semantic information to that token, allowing the model to recall the memorized facts when queried with semantically related prompts.

**Evaluation.** To assess whether the model truly generalizes, we evaluate its performance across 3 increasingly challenging settings. These three generalization settings are also illustrated in Figure 1.

(a) *Unseen associations*: Can the model retrieve facts excluded from Phase-2 using the training prompts? This tests whether it learned the underlying relation rather than just memorizing examples.

(b) *Unseen prompts*: Can the model retrieve all facts using new prompts that are semantically similar to the training prompt in Phase-2? Our goal is to evaluate whether the model has internalized the semantics of the training prompt and can generalize beyond exact-match training prompts.

(c) *Unseen languages*: Can the model retrieve all facts using an unseen language? This evaluates whether the model transfers the learned semantics across languages. For a pre-trained multi-lingual LLM, if the model truly understands the semantics, it should be able to recognize the relation and extract the fact in other languages.

**Dataset.** To ensure novelty and avoid contamination from pre-trained knowledge, we construct a fully synthetic dataset. Synthetic data eliminates confounding factors present in existing knowledge, providing a clean setup to study how models acquire new information. Each fact is represented as a subject–relation–object triplet, a standard abstraction in knowledge graphs (e.g., Wikidata (Vrandečić & Krötzsch, 2014)) and prior work on factual knowledge in LLMs (Petroni et al., 2019; Yao et al., 2025; Du et al., 2025; Hu et al.). Beyond isolated triplets, we test generalization to more complex structures by composing two triplets (e.g., training with A→B and B→C, then evaluating A→C) to probe multi-hop reasoning. Our dataset covers five T-REx (Elsahar et al., 2018) relations: *author*, *capital*, *educated at*, *genre*, and *mother*. The relation *mother* is further linked to *educated at* (the object of *mother* becomes the subject of *educated at*) to enable triplet composition experiments. For each relation, we prompt GPT-4 (gpt-4-turbo-2024-04-09) with representative T-REx examples and generate 100 fictional subject–object pairs, each paired with 100 alternative objects for multiple-choice evaluation. We additionally create 20 diverse natural language prompts per relation (10 for training, 10 for testing), along with 3 unrelated prompts for out-of-domain evaluation. To assess cross-lingual robustness, all prompts are translated into German, Spanish, Chinese, and Japanese. Detailed generation settings and examples are provided in Appendices C.2 and C.3.

**Evaluation Metrics.** We evaluate the output of a model using three metrics: (1) *Generation accuracy*, where the model generates 50 tokens under greedy sampling. We check whether the generated output contains an exact match of the target object; (2) *Multiple-choice accuracy*, where the model must select the correct answer from a list of 100 candidate options per fact; (3) *Probability* assigned by the model to the object. For example, given the input "The capital of Germany is", we take the probability for predicting the token "Berlin". For multi-token objects, we compute the joint probability by multiplying the probabilities of each token. The absolute probability reflects how likely the model is to generate the desired answer independently, without relying on generation-specific hyperparameters. The formal definitions can be found in Appendix B.

# 4 INVESTIGATING THE EFFECTIVENESS OF MEMORIZE-THEN-GENERALIZE

In this section, we ask the question – *Can LLMs generalize effectively from memorized data*? We investigate it using the Qwen2.5-1.5B model and find that it can indeed generalize effectively. We further show that the finding holds consistently across 8 models (ranging from 1B to 14B) spanning 4 different families (Qwem2.5, Llama2, Llama3.2, and Phi-4), including both base and instruction models. A full list of models is provided in Appendix C.

We apply our two-stage framework to our synthetic dataset. In Phase-1, we train for 20 epochs using different key tokens across relations, with facts in the same relation sharing a token. After it, the model gets an average generation accuracy of 0.36 across all test prompts. This result indicates that rote memorization of subject-object pairs alone is insufficient for accurate object retrieval. We therefore proceed to Phase-2 and evaluate our method under three generalization settings as mentioned in Section 3.

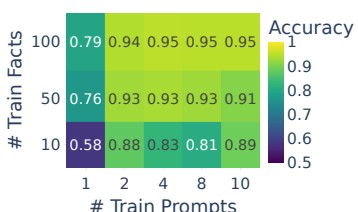

Figure 2: **Effective generalization with minimal training, facts, and prompts.** In Phase-1, the model rote-learns 100 facts per relation using a synthetic key token for 20 epochs, achieving an accuracy of 0.36. In Phase-2, the model is fine-tuned for 1 epoch while varying the number of training prompts (x-axis) and memorized associations (y-axis). Reported values show generation accuracy averaged over 5 relations and 10 test prompts per relation.

**LLMs can generalize to held-out facts and prompt variants.** As shown in Figure 2, after just one epoch of Phase-2 training—using only 50 memorized associations (*y-axis*) and a single training prompt (*x-axis*)–the model achieves a generation accuracy of 0.76 on unseen facts. This demonstrates that the model can generalize beyond rote memorization. Intuitively, increasing either the number of prompts or the amount of training data in Phase-2 should further enhance generalization. We systematically vary the number of training prompts and associations during this phase in Figure 2 and observe that the model generalizes robustly across a wide range of settings.

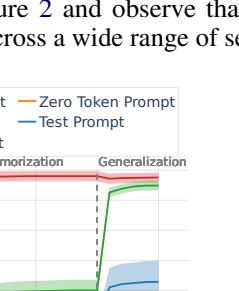

Figure 3: **LLMs generalize to held-out facts and unseen prompts.** Phase-1: Epochs 1–20; Phase-2: Epochs 21–25. Results are averaged over 5 relations, each with 1 training prompt, 3 unrelated prompts, and 10 test prompts.

To better understand the surprising generalization performance, we analyze training dynamics through object probabilities. As shown in Figure 3, after Phase-1 the model assigns high probability to the correct object only when provided with the exact key token, e.g. `Gene Finley [X]`. In contrast, when given only the subject (zero token prompt), e.g. `Gene Finley`, the probability is near zero. This suggests that memorization relies on the key token as an anchor for the relation, since the model cannot predict the object from the subject alone.

We then continue the Phase-2 using a semantically meaningful training prompt $\mathcal{P}_r^{train}$ on $k$ facts. We evaluate whether the model can: (a) generalize to retrieve the remaining $n - k$ memorized associations using $\mathcal{P}_r^{train}$, and (b) further generalize to all $n$ facts when prompted with semantically equivalent variants $\mathcal{P}_r^{test}$. After just one epoch, the model's object probability on held-out facts with $\mathcal{P}_r^{train}$ jumps from 0.18 to 0.79. For $\mathcal{P}_r^{test}$ variants, it increases from 0 to 0.17. Crucially, performance remains unchanged for zero token and unrelated prompts, confirming that the model has learned the semantic meaning of the key token, not merely subject–object patterns. Similar gains are observed in other metrics (Figure 8).

**LLMs generalize across languages.** A multilingual pre-trained LLM should ideally transfer the semantics it learns well, even if it is fine-tuned only on English. As shown in Figure 4, the model achieves strong generation accuracy across German, Spanish, Chinese, and Japanese. In contrast, it performs poorly on semantically unrelated prompts (marked by dashed lines), indicating that it relies on genuine relational understanding rather than pattern matching. Figure 12 further shows a clear ranking in object probabilities by language: English leads, followed by Spanish, German, Japanese, and Chinese. This indicates that while the model exhibits some cross-lingual semantic generalization,

| Rote Memorization | | | | Generalization | | | | |
|---|---|---|---|---|---|---|---|---|
| | Key Token Prompt | | | | Train Prompt | | Test Prompt | |
| Epoch | Acc | Prob | $k$ | Epoch | Acc | Prob | Acc | Prob |
| 3 | 0.48 | 0.12 | 50 | 1 | 0.38 | 0.13 | 0.35 | 0.076 |
| 6 | 1.00 | 0.94 | 50 | 1 | 0.94 | 0.60 | 0.89 | 0.41 |
| 10 | 1.00 | 1.00 | 50 | 1 | 0.94 | 0.69 | 0.98 | 0.62 |
| 20 | 1.00 | 1.00 | 50 | 1 | **1.00** | **0.85** | **0.98** | **0.69** |
| 10 | 1.00 | 1.00 | 1 | 8 | 1.00 | 0.68 | 0.75 | 0.35 |
| 20 | 1.00 | 1.00 | 1 | 8 | **1.00** | **0.70** | **0.76** | **0.36** |

Table 1: **(a) Memorize more, generalize better. (b) One fact and one prompt are enough to generalize.** Phase-1: the model memorizes 100 `author` facts. Phase-2: fine-tuning from different Phase-1 checkpoints with a single prompt, we evaluate accuracy and object probability while varying Phase-2 dataset size ($k$). Results generalize across other relations (Table 11) and models (Table 6).

it performs better on languages that are more similar to the training language. This hypothesis is also supported by the representation analysis in Section 5.

**The memorize-then-generalize phenomenon is robust across different models.** To test whether our findings extend beyond a single model, we test using 8 models: Qwen2.5-1.5B, Qwen2.5-7B, Qwen2.5-14B, Qwen2.5-1.5B-Instruct, Qwen2.5-14B-Instruct (Qwen, 2024), LLaMA2-7B (Touvron et al., 2023), LLaMA3.2-1B (Grattafiori et al., 2024), and Phi-4 (Abdin et al., 2024). Model details are listed in Table 4. We fix a challenging configuration, $k = 50$ and $|\mathcal{P}_r^{train}| = 1$, where generalization is particularly difficult (see Figure 2). As shown in Figure 11, all models improve substantially after Phase-2 across three metrics, demonstrating that generalization beyond memorized data is a robust, transferable capability across model families and scales.

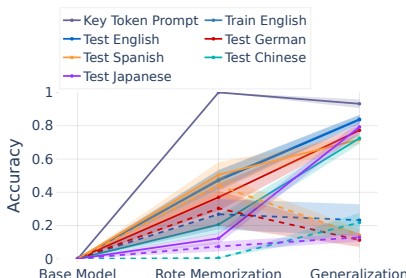

Figure 4: **Generalization with multilingual prompts.** Phase-1: memorize 100 facts per relation. Phase-2: fine-tune on 50 facts and 10 English prompts per relation. We report the generation accuracy averaged over 5 relations. Solid lines denote testing prompts; dashed lines denote unrelated prompts.

Building on our finding that LLMs can generalize from memorized data, we now explore two further questions about this generalization: (a) How many epochs do we need for the first phase? (b) How many examples are needed in the second phase for the model to align the semantics of the key token?

**(a) Memorize more, generalize better.** Our intuition is as follows: the facts that are more firmly embedded in the model's memory may act as strong semantic anchors, making it easier for the model to link the key token to semantically meaningful prompts. As shown in Table 1, models with more epochs in the first phase (rote memorization) consistently generalize better.

**(b) One fact and one prompt are enough to generalize.** Contrary to the common belief that generalization requires diverse prompts, our results show that the model is able to generalize effectively from just a single well-memorized association paired with one training prompt (see Table 1). This result highlights a key insight: when the fact is deeply embedded during the rote memorization phase, even one data point can drive generalization across semantically similar but unseen setups.

To probe the role of the key token in memorize-then-generalize, we ran ablations (Appendix D). Results show: (1) the model cannot generalize by memorizing subject–object pairs alone (Figure 9); and (2) while it can generalize to a new meaningful prompt (train-2) after memorizing another (train-1) (Figure 10), Phase-2 training substantially alters performance on train-1. These findings suggest the synthetic meaningless key token acts as a crucial anchor for repurposing relational knowledge. A significance test (Appendix E) confirms these effects are statistically robust.

## 5 Understanding Training Dynamics through Representations

In Section 4, we observe that the model learns subject–object associations using the key token prompt in Phase-1 training. . In Phase-2, it generalizes these associations to semantically meaningful prompts.

A natural hypothesis about why the generalization happened is that the model aligns the semantics of the synthetic key token with the semantically meaningful prompts. We therefore want to analyze the model's internal representations to understand the alignment. To test this, we observe how the representation of key-token prompts changes over training. Specifically, we focus on prompts of the form `Subject [X]`, where `[X]` is a relation-specific synthetic key token.

Since autoregressive LLMs use the last-layer representation of the input's final token to predict the next token, we take this hidden state as the representation of an input. To study how the model learns during training, we use two complementary analyses. First, we examine the clustering of representations, which reflects *how the model organizes relation-specific structures*. Second, we measure the similarity between key-token prompts and semantically meaningful prompts, which captures *how the model assigns semantics to the synthetic key token*. Together, these methods provide complementary perspectives on the dynamics of bridging the gap between key token and meaningful prompts. Further implementation details are provided in Appendix F.1.

**The model acquires relational structure through rote learning in Phase-1, which is further reinforced during Phase-2 fine-tuning.** We visualize the embeddings using PCA (Maćkiewicz & Ratajczak, 1993) (details in Appendix F.2). To move beyond qualitative visualization, we introduce the $\Delta$CosSim metric, defined as the average cosine similarity *difference between intra- and inter-cluster pairs* (formal definition in Appendix F.3). A higher $\Delta$CosSim indicates that embeddings within the same cluster are more tightly grouped (high intra-cluster similarity) and embeddings across different clusters are more distinct (low inter-cluster similarity). This difference directly captures the separation between clusters. As shown in Figure 5, the base model exhibits highly entangled representations, with a low $\Delta$CosSim of 0.058, reflecting a lack of relational structure. As rote memorization progresses, clusters become increasingly separated, with $\Delta$CosSim rising to 0.116 at epoch 2 and 0.191 at epoch 20, indicating that the model begins to differ-

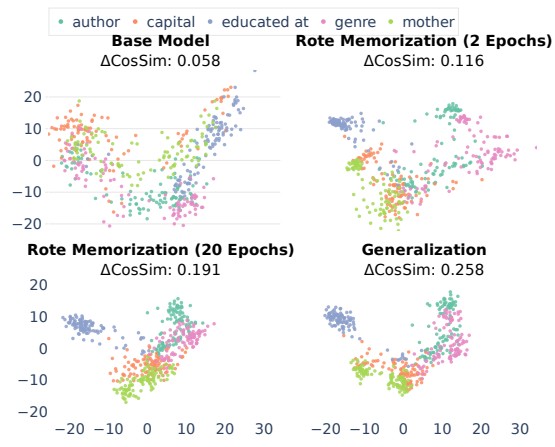

Figure 5: **Later-stage checkpoints from our training can better encode structural relational knowledge.** Qwen2.5-1.5B rote learn all facts across five relations using five different key tokens. Phase-2 fine-tuning was conducted with $k = 50$ examples and $|\mathcal{P}_r^{train}| = 1$ per relation, fine-tuned for one epoch.

entiate relations through memorization. After Phase-2 fine-tuning, the clusters become most distinct, with $\Delta$CosSim further increasing to 0.258, revealing a clear separation of different relations.

| Prompt | Phase-1 (Rote Memorization) | Phase-2 (Generalization) |
|---|---|---|
| Train | $0.87 \pm 0.01$ | $0.90 \pm 0.01$ |
| Test | $0.58 \pm 0.01$ | $0.71 \pm 0.02$ |
| Unrelated | $0.50 \pm 0.01$ | $0.50 \pm 0.02$ |

Table 2: **Key token alignment with desired semantics after Phase-2.** Cosine similarity of the key token's representation with semantically meaningful prompts (Train, Test) and unrelated prompts for the same models as in Figure 5.

**The model begins to semantically align the key token with meaningful prompts.** We compute the cosine similarity of the key token with meaningful prompts. As shown in Table 2, similarity with semantically related prompts (*Train* and *Test*) increases substantially after Phase-2 fine-tuning. This supports the hypothesis that the key token is reinterpreted to the desired semantics. One interesting finding is that the alignment with *unrelated* prompt does not change, further confirming our hypothesis. Figure 16 shows more pairwise alignments. Figure 17 shows the same effect at the per-relation level. Furthermore, the model retains its understanding of the key token and continues to generalize when learning additional facts (Figure 19).

**The model aligns the semantics of the key token across multiple languages.** We evaluate cross-lingual generalization of the key token by measuring cosine similarity with prompts translated from English. After Phase-2 fine-tuning, alignment increases across languages in the order of Spanish, German, Japanese, Chinese (Figure 18), revealing cross-lingual

retrieval accuracy and object probability(Figure 12). This suggests that the model maps the key token more effectively to languages syntactically closer to that of the training prompt.

**The two-phase training does not affect unrelated knowledge.** We also examine how existing knowledge in LLMs interacts with the learning of new facts in our framework. We categorize existing knowledge into two types: (i) *related knowledge*, overlapping with the five injected relations, and (ii) *unrelated knowledge*, randomly sampled from the T-REx dataset (Elsahar et al., 2018). Our hypothesis is that Phase-1 memorization with synthetic subject–object pairs and a key token provides no meaningful representation of the new relation, and Phase-2 fine-tuning reinterprets the key token specifically toward the target relation's semantics. Thus, unrelated relations should remain unaffected. Our results confirm this: both generation accuracy (Table 8, 9, 10) and representation analysis (Figure 20 to 25) show that performance on related knowledge is influenced by training, while that on unrelated knowledge is preserved. By relying on synthetic triplets, we establish a clean experimental setup that enables controlled study of knowledge retention and forgetting in continual learning. Full results for both related and unrelated knowledge are provided in Appendix F.6.

## 6 APPLICATIONS OF THE MEMORIZE-THEN-GENERALIZE FRAMEWORK

We find that LLMs can generalize from rote-memorized data has both positive and negative implications. On the positive side, it provides a more efficient path to knowledge injection and suggests that rote memorization can even enhance reasoning performance over factual knowledge. On the negative side, the same mechanism creates risks if memorized data is repurposed for harmful generations.

### 6.1 MORE EFFECTIVE AND EFFICIENT KNOWLEDGE INJECTION

A central implication of our results is that rote memorization offers a more efficient and effective way to learn new facts. By first encoding subject–object pairs through a single synthetic key token and then mapping this token to semantic prompts, our method enables efficient knowledge injection with substantially fewer computational and data resources. Because we rely on only one synthetic key token, the model can learn to associate and generalize factual relationships without the need to retrain on large corpora. Moreover, training on a small subset of memorized facts with meaningful prompts is sufficient to achieve strong generalization, and we demonstrate that even a single well-chosen prompt can effectively transfer factual knowledge to unseen cases.

**Comparison with SFT.** In supervised fine-tuning (SFT), the model is directly trained on meaningful training prompts ($\mathcal{P}_r^{train}$), which are typically $20\times$ longer than the single-token key token used in our framework. As shown in Figure 6a, this makes our method more effective in low-data regimes and more efficient overall. With just one training prompt, our method achieves far higher accuracy than SFT with the same number of training tokens. At ten training prompts, both reach $\sim$0.9 accuracy, but ours does so with half the tokens. The key efficiency and performance gain comes from reusing the same key token across facts, while SFT must repeatedly fine-tune on full-length prompts. Full details of the training, evaluation, and the training tokens are provided in Appendix G.3 and G.1.

**Comparison with ICL.** We also compare our method against in-context learning (ICL), where the fact is provided directly as part of the input. Even in this idealized setup, our framework consistently outperforms ICL (Figure 6b), producing higher object probabilities and exhibiting a clearer separation between related and unrelated prompts. In contrast, under the ICL setting, the model tends to assign similarly high probabilities to both related and unrelated prompts, which is not expected. Furthermore, our approach demonstrates substantially lower variance across different languages and prompt formulations (Figure 26), indicating that the injected knowledge is not only captured more efficiently but also generalized more systematically. These findings suggest that, unlike ICL—which relies on temporary contextual cues—our method enables the model to internalize and retain factual associations within its latent representations, leading to more stable knowledge integration.

### 6.2 REASONING PERFORMANCE BENEFITS FROM ROTE MEMORIZATION

Reasoning over facts often requires more than direct recall. Two common settings are *reversal reasoning*—answering the inverse of a known fact, and *multi-hop reasoning*—combining multiple facts to infer new ones. These tasks typically challenge LLMs, as they require structured manipu-

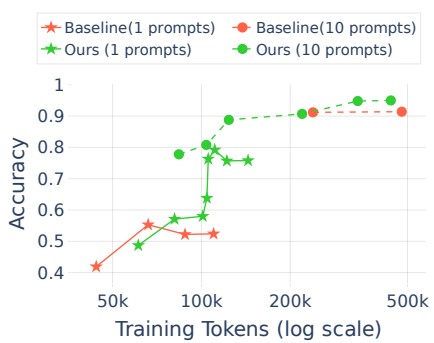

(a) Memorize-then-generalize enables more efficient fact learning than SFT.

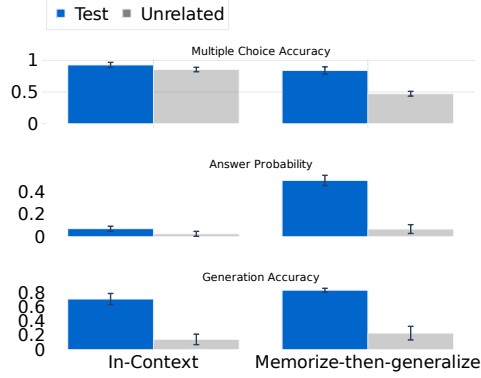

(b) Memorize-then-generalize enables more effective fact learning than ICL.

Figure 6: **Comparison of Memorize-then-generalize performance with SFT and ICL.** (a) Ours vs. SFT. (b) Ours vs. ICL. Base model: Qwen2.5-1.5B, all the results are averaged across 5 relations and 10 prompts per relation.

| Rote Memorization (R1 only) | Generalization (R1 + R2) | | |
|---|---|---|---|
| Epoch | Epoch | Generation Accuracy | Multiple-Choice Accuracy |
| 0 (No rote memorization) | 15 | 0.14 | 0.08 |
| 5 | 15 | 0.14 | 0.14 |
| 10 | 15 | 0.14 | 0.13 |
| 15 | 15 | 0.14 | 0.16 |
| **20** | **15** | **0.36** | **0.16** |

Table 3: Impact of rote memorization depth (R1) on generalization performance with composed relations (R1+R2). Longer training on R1 leads to higher accuracy on the inference task.

lation of stored knowledge rather than surface-level memorization. Surprisingly, we find that rote memorization of atomic facts improves performance on both tasks.

**Reversal reasoning.** Given a memorized fact such as "X is the mother of Y," reversal reasoning requires answering "Who is the child of Y?" Prior work shows that supervised fine-tuning (SFT) typically fails unless explicitly trained with reversal examples (Berglund et al., 2023; Allen-Zhu & Li, 2023; Golovneva et al., 2024). In contrast, under memorize-then-generalize, we first memorize the forward relation (A→B) and then train on the reverse relation (B→A) for only a subset of the memorized facts. The model is then able to generalize this reversal to unseen facts during Phase-2. Moreover, repeating the rote memorization phase strengthens this effect. For example, in Qwen2.5-1.5B, accuracy on reversal queries for the "mother" relation increases from 0 to 0.26 after Phase-2 training (Figure 27), with more epochs of memorization further boosting generalization.

For both the reversal reasoning, with our SFT baseline, the model got an accuracy of 0.01. For the ICL baseline, where we add the original facts just before the question, we got an accuracy of 1. These results show that simple SFT fails to support the reversal and multi-hop reasoning, while ICL can solve the task.

**2-hop reasoning.** This setting requires composing related facts. For example, if A is the mother of B and B is educated at C, the model should infer that the child of A is educated at C. We first train the model to memorize one relation ("A is the mother of B"), then introduce a second relation ("B is educated at C"). Importantly, A and C never co-occur in a training sequence. As shown in Table 13, deeper memorization of the first relation significantly improves performance on the composed inference, with A→C accuracy rising from 0.14 (no memorization) to 0.36. These results indicate that rote memorization provides a stable substrate that the model can later leverage for reasoning. Far from being shallow, memorization strengthens the factual scaffolding needed for more complex inference. For the ICL baseline, the model can get an accuracy of 0.95 when we have the two facts in the context. We provide all the details in the Appendix G.6.

## 6.3 IMPLANTING HARMFUL FACTS USING MEMORIZATION

While efficient knowledge injection is desirable, the same mechanism also presents potential risks if misused. Specifically, once factual associations are memorized, the model can be steered to generalize them into harmful or unintended contexts through minimal additional fine-tuning. For example, after a model has memorized 100 benign relational facts of the form "A is the mother of B," fine-tuning on 50 malicious variants (e.g., statements expressing harmful or adversarial relations) causes the model to propagate this undesirable association to the remaining 50 pairs. In effect, the attacker can corrupt the model's generalization behavior while the model still provides correct answers to benign prompts, making such manipulation difficult to detect from casual inspection (Figure 7).

This finding is notable for two key reasons. First, it reveals that the memorize-then-generalize behavior can give rise to what we term *dual generalization*, wherein both benign and harmful interpretations coexist within the model's representations. In other words, the model retains its original factual knowledge but simultaneously acquires the capacity to apply it in malicious or undesirable ways. Second, this behavior exposes a concrete security vulnerability: adversaries could exploit such mechanisms to weaponize memorized knowledge from pretraining, repurposing it for harmful outputs without overwriting or visibly corrupting the model's apparent capabilities. For example, a compromised model might respond correctly with "A is the mother of B" in normal contexts but also produce "A is abusing B" when prompted in slightly altered conditions. This duality underscores the importance of developing safeguards and interpretability tools to detect and mitigate malicious fine-tuning. Full experimental details and additional analyses are provided in Appendix H.

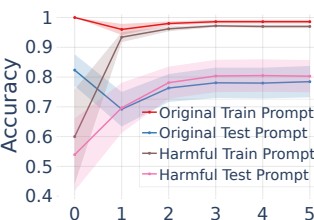

Figure 7: **Memorized facts can be repurposed harmfully.** Model correctly answers the malicious variant ('A is abusing B') while also answering the original benign relation correctly.

## 7 CONCLUSION

We introduce a *memorize-then-generalize* framework to better understand how LLMs memorize and generalize on the testbed of factual knowledge. In doing so, we uncover an intriguing phenomenon: **LLMs can generalize over their own memorized data**, demonstrating that memorization does not necessarily imply unrecoverable overfitting. Instead, rote memorized data can serve as a foundation for abstraction and reasoning, allowing models to extend and reinterpret the learned patterns. It suggests that memorization and generalization are not opposing forces but complementary processes that jointly enable knowledge acquisition.

**This finding opens up several practical opportunities.** By leveraging the natural ability of LLMs to generalize from memorized data, we can design efficient and controllable knowledge injection methods that require far fewer samples and computational resources than traditional SFT and ICL. Moreover, the same mechanism supports further higher-level reasoning. However, it also calls attention to new risks: memorized knowledge can be repurposed or "weaponized" through malicious fine-tuning on a small amount of memorized data, resulting in harmful generalization without overtly disrupting benign capabilities.

**An important direction for future research is to examine whether the memorize-then-generalize dynamics extend to other domains that intertwine factual recall and abstraction**, such as mathematical problem solving, scientific reasoning, and code generation, where memorized patterns may similarly serve as scaffolds for higher-level generalization. Ultimately, understanding the interplay between memory and generalization may enable a new class of training paradigms that purposefully cultivate "structured memorization" as a foundation for robust abstraction, interpretability, and alignment. Such work would not only deepen our scientific understanding of LLM cognition but also inform the design of safer and more efficient systems capable of learning and reasoning.

## ETHICS STATEMENT

This work explores how memorization in LLMs can support generalization. While this offers benefits for knowledge injection and reasoning, it also introduces risks: memorized facts, including those from pre-training, can be repurposed for harmful generations when exposed to malicious supervision. To reduce misuse potential, we restrict harmful examples to controlled, synthetic cases and do not release any sensitive data. We highlight these risks to encourage future research on detection, mitigation, and safer use of memorization in LLMs.

## REPRODUCIBILITY STATEMENT

To ensure the reproducibility of our results, we will make all data, code, and execution environments publicly available upon acceptance. The dataset is also included as supplementary material, and a detailed description of our LLM training and inference setup is provided in Appendix C, G.3, and F.1.

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

## A  LLM USAGE

In this paper, we use LLMs in those parts:

1. Synthetic data generation, while we're generating our synthetic dataset, we use the examples from T-Rex (Elsahar et al., 2018) dataset and then prompt GPT-4 to generate similar but synthetic data for us. We describe the generation details in both the main paper in Section 3 and the Appendix C.2.2.

2. We use the ChatGPT online platform to suggest transition words and rephrase our writing for improved readability. We also use ChatGPT to generate tables in the correct LaTeX format; however, all results are manually verified.

3. LLM-based copilots assisted in generating some portions of code; all the code is verified by the authors to ensure correctness.

4. We use AI2 Paper Finder **??** and OpenAI Deep Research [2] to help with the identification of relevant literature.

## B  EXPERIMENTAL SETUPS AND EVALUATION

**Evaluation Metrics.**   We evaluate the output of a model $\theta$ given an input $p(s)$ using three methods: (1) the *absolute probability* assigned by the model to the correct answer $o$; (2) a *multiple-choice* setting, where the model must select the correct answer from a list of 100 candidate options per fact in our dataset; (3) open-ended generation, where the model freely generates text based on the input, and we check whether the generated output contains an exact match of the target object $o$. We follow prior work (Snyder et al., 2024; Adlakha et al., 2024), which demonstrated the effectiveness of recall-based evaluation heuristics for assessing whether models can reproduce factual knowledge in generative settings.

We compute the *object probability* over multiple tokens as follows:

$$P_\theta(o \mid p(s)) = P_\theta(o^{(1)} \mid p(s)) \cdot \prod_{i=2}^{|o|} P_\theta(o^{(i)} \mid o^{(1)}, \ldots, o^{(i-1)}, p(s)) \tag{1}$$

where $|o|$ denotes the number of tokens in $o$, and $P_\theta(o^{(i)} \mid o^{(1)}, \ldots, o^{(i-1)}, p(s))$ is the conditional probability of predicting the $i$-th token $o^{(i)}$ of $o$ given its preceding tokens and the prefix $p(s)$.

For the multiple-choice question, to determine whether model $\theta$ can retrieve a fact $f = \langle s, r, o^* \rangle$, we test whether given an input $p(s)$, $\theta$ can choose the correct object $o^*$ from among a set of $M$ unique

---

[2]https://chatgpt.com/

alternatives. Specifically, given fact $f$, we redefine it as $f = \langle s, r, o^*, \mathcal{O} \rangle$, where $\mathcal{O}$ is a set of $M$ plausible but incorrect alternatives.

$$\text{pred}_\theta(f) \triangleq \underset{o \in \{o^*\} \cup \mathcal{O}}{\text{argmax}} \; P_\theta(o \mid p(s)) \tag{2}$$

denotes the prediction of $\theta$ for the fact $f = \langle s, r, o^*, \mathcal{O} \rangle$.

The predicted object has the maximal object probability within $\{o^*\} \cup \mathcal{O}$.

For the *open-ended generation*. Given a fact $f = \langle s, r, o^* \rangle$ and a model $\theta$, we provide the input $p(s, r)$ to the model and let it generate for $k$ tokens $t_1, t_2, ..t_k$. We consider the answer to be correct if $y^* \subseteq \{t_1, t_2, ..., t_k\}$ leading to the prediction $pred_\theta(f) = y^*$.

We evaluate the factual knowledge of model $\theta$ over a test dataset $\mathcal{D}_r^{test} = \{f_i\}_{i=1}^m$ using accuracy as a metric for both the response test and multiple-choice test:

$$\text{acc}(\theta, \mathcal{D}_r^{test}) \triangleq \frac{\sum_{f \in \mathcal{D}} \delta \left( o^* = \text{pred}_\theta(f) \right)}{|\mathcal{D}|} \tag{3}$$

where $\delta(\cdot)$ is the indicator function.

## C    REPRODUCIBILITY

In this section, we provide the base model we're using, the dataset generation details, the training and testing prompts generation details, the training implementation and hyperparameters, and the evaluation details.

### C.1    BASE MODELS

We show the details of the base model we used in this paper in Table 4.

| Model | Link |
|---|---|
| Qwen2.5-1.5B | https://huggingface.co/Qwen/Qwen2.5-1.5B |
| Qwen2.5-1.5B-Instruct | https://huggingface.co/Qwen/Qwen2.5-1.5B-Instruct |
| Qwen2.5-7B | https://huggingface.co/Qwen/Qwen2.5-7B |
| Qwen2.5-14B | https://huggingface.co/Qwen/Qwen2.5-14B |
| Qwen2.5-14B-Instruct | https://huggingface.co/Qwen/Qwen2.5-14B-Instruct |
| Llama2-7B | https://huggingface.co/meta-llama/Llama-2-7b |
| Llama3.2-1B | https://huggingface.co/meta-llama/Llama-3.2-1B |
| Phi-4 (14.7B) | https://huggingface.co/microsoft/phi-4 |

Table 4: **Base models and their download links used in this paper.**

### C.2    SYNTHETIC DATASET

In this section, we provide the details of generating the synthetic dataset and some examples of our synthetic dataset. All the data are generated through the GPT-4 API: gpt-4-turbo-2024-04-09. In all the generations, we set the temperature as 0.7, and use the default number for other generation parameters.

To study model generalization on factual knowledge, we construct a synthetic dataset of fictional (subject, object) pairs for a given relation (e.g., educated_at). This dataset is generated using a two-phase pipeline powered by the OpenAI API. Our goal is to create realistic-looking but fictional entities and use them to form factual statements, along with high-quality distractors for multiple-choice evaluation.

### C.2.1    PROMPTING FOR GPT-4

The generation process begins by loading example entities from the T-REx dataset corresponding to the target relation. These examples serve as demonstrations to guide the LLM's generation. For

each entity type, we construct a prompt that asks the LLM to produce a list of similar but fictional entities. We emphasize in the prompt that the entities should be novel—i.e., not drawn from the model's training data or the real world. For instance, when generating synthetic universities, the prompt looks like:

```
system prompt = "You are an expert to come up with totally
new entities." user prompt = f"""Generate a list of 20 synthetic
entities for the entity university, which should look similar to the
following examples: 1. Harvard University 2. Stanford University
3. Massachusetts Institute of Technology The synthetic entities
should be unique and unknown to you. Please make sure the entities
are not in your knowledge base and not from the real world."""
```

The model returns a list of synthetic subject entities, which we parse and clean. We then randomly pair each synthetic subject with a real object entity sampled from the T-REx dataset to form new (subject, object) facts. Although the objects are real, the facts themselves are synthetic, since these subject-object pairs do not occur in the real world and introduce novel associations.

To support multiple-choice evaluation, we also generate 99 distractor objects per fact by sampling from a pool of real object entities. We ensure that these distractors are unique, unrelated to the true object, and do not share substrings with each other.

This synthetic dataset allows us to precisely control for memorization and test the model's ability to generalize across prompts and entities it has never seen before. We provide the full dataset in the supplementary materials.

### C.2.2 DATASET EXAMPLES

Here we provide one example for each of the relations in Table 5.

Table 5: Example synthetic facts constructed for various relations. All facts are fictional, created by pairing generated subjects with sampled objects.

| Relation | Subject (Generated) | Object (Sampled) |
|---|---|---|
| Author | Symphony of the Forsaken | Joseph Boyden |
| Instance of | Blazepeak | Astronomical Observatory |
| Educated at | Clara Bellmont | Redwood University |
| Capital | Kalindor | Nowy Targ |
| Mother | Countess Genevieve Lorne | Giselle Harper |

As one alternative facts example of the first fact:

```
'lutheran', 'jan guillou', 'virginia woolf', 'lorenz hart',
'stephen hillenburg', 'helen bannerman', 'mervyn peake', 'neutron
star', 'brian azzarello', 'achdiat karta mihardja', 'ivan
turgenev', 'marion zimmer bradley', 'thomas middleton', 'bill
gates', 'edgar', 'jonah', 'philippa gregory', 'carlo collodi',
'vaidyanatha dikshita', 'hesiod', 'johannes kepler', 'pope
gregory x', 'christina crawford', 'kalki krishnamurthy', 'saxo
grammaticus', 'daniel defoe', 'hume', 'herman wouk', 'eiichiro
oda', 'lois mcmaster bujold', 'lee child', 'koushun takami',
'schumann', 'william gibson', 'lynn okamoto', 'pope pius ix',
'ai yazawa', 'clare boothe luce', 'hippocrates', 'plotinus',
'alexander hamilton', 'ambrose', 'leslie charteris', 'sakyo
komatsu', 'pierre choderlos de laclos', 'jude watson', 'the
prophet', 'justinian i', 'james ivory', 'thomas mann', 'trenton
lee stewart', 'steele rudd', 'pran', 'john ruskin', 'brian
lumley', 'jacqueline rayner', 'evan hunter', 'gilles deleuze',
'michael lewis', 'jane austen', 'jimmy wales', 'christos tsiolkas',
'candace bushnell', 'alexander glazunov', 'the pittsburgh cycle',
'hermann hesse', 'mamoru oshii', 'germaine greer', 'samuel taylor
```

```
coleridge', 'amish tripathi', 'pope boniface viii', 'julius
caesar', 'irvine welsh', 'max weber', 'jules verne', 'jeff lynne',
'mary wollstonecraft shelley', 'johann wolfgang goethe', 'jan de
hartog', 'abraham lincoln', 'feynman', 'ernest raymond', 'lao tzu',
'eudora welty', 'hiro mashima', 'nikephoros phokas', 'murasaki
shikibu', 'bruce sterling', 'peter lombard', 'marshall mcluhan',
'garth nix', 'anton szandor lavey', 'quintus smyrnaeus', 'william
gaddis', 'patricia highsmith', 'martin caidin', 'jack london',
'allan sherman', 'armijn pane'
```

## C.3 TRAINING AND TESTING PROMPTS

To generate the different training and testing prompts, the authors wrote one base prompt for each relation, which is every Train-1 in Appendix C.3. For each relation, we begin with the base prompt template. For example, for the relation educated at, the base prompt is:

```
{head} is educated at
```

We use GPT-4 to generate multiple semantically equivalent versions of the base prompt. The model is instructed to:

- Generate $N$ variants (typically $N = 20$),
- Maintain the original semantic meaning,
- Vary the vocabulary and sentence structure,
- Produce prompts with increasing complexity, ranging from simple to complex (as measured by readability scores).

The prompt we're using for GPT-4:

```
 system prompt = "You are an expert in linguistics and prompt
engineering." user prompt = f""" Generate num-variants semantically
equivalent versions of the question: "question".  You should
have those variants from very simple one to very complex one.
For the very complex one, you can use more complex grammar and
vocabulary which can achieve Flesch Reading Ease score of 30 or
lower. Use progressively more complex grammar and vocabulary. Do
not include the number of variants in the output. Do not include
any explanations or additional text.  Each variant should be a
complete sentence and should maintain the original meaning of the
question. Please ensure that the variants are distinct from each
other and from the original question. Please try to not repeat the
same sentence structure or vocabulary in the
```

**Train/Test Split:** The original base prompt is always included in the training set. In addition, 9 semantically diverse variants are randomly sampled to form the rest of the training set. The remaining variants are used as the test set. Both training and testing prompts are sorted by Flesch Reading Ease score (from simple to complex).

This process allows us to systematically test whether models can generalize retrieval across prompts that vary in phrasing and complexity, even when the underlying relation remains the same.

### C.3.1 PROMPTS FOR EACH RELATION

The unrelated prompts are the same for all relations, which is some random token prompt:

- Unrelated-1: {subject} Hi! How are you doing today? Do you have any plans for the weekend? I hope you are having a great day!
- Unrelated-2: {subject} How is the weather in your area right now? Do you think it will change later? I hope you are staying warm and dry!

- Unrelated-3: {subject} What is your favorite color? Do you have any specific reason for liking it? I hope you find it beautiful and calming!

**Relation 1: authors**

- Train-1: The author of {subject} is
- Train-2: Do you know who penned {subject}?
- Train-3: Who is the scribe behind {subject}?
- Train-4: The writer of the masterpiece, {subject}, is who?
- Train-5: The literary work known as {subject} was written by whom?
- Train-6: Can you reveal the identity of the person who composed {subject}?
- Train-7: Can you disclose the name of the individual who scripted {subject}?
- Train-8: Can you identify the person who authored {subject}?
- Train-9: Could you elucidate who the creator of {subject} is?
- Train-10: The literary opus, {subject}, can be attributed to which individual?

- Test-1: Who wrote {subject}?
- Test-2: Can you tell me who the author of {subject} is?
- Test-3: The one who breathed life into the work known as {subject} is?
- Test-4: Who was the one to weave words into the creation known as {subject}?
- Test-5: The person who crafted {subject} is?
- Test-6: The written piece {subject} was the brainchild of which writer?
- Test-7: Who should receive credit for the authorship of {subject}?
- Test-8: The written work {subject} is credited to which writer?
- Test-9: Who holds the distinction of being the author of {subject}?
- Test-10: Who is the individual that wrote {subject}?

**Relation 2: instance of**

- Train-1: {subject} is an instance of
- Train-2: {subject} is a case of what?
- Train-3: What form or type does {subject} pertain to?
- Train-4: What unique genre or form does {subject} serve as a representation of?
- Train-5: In what classification does {subject} belong?
- Train-6: Could you determine the precise class that {subject} epitomizes?
- Train-7: What distinct genre or classification does {subject} echo?
- Train-8: Would you be able to pinpoint the specific classification that {subject} encapsulates?
- Train-9: Can you ascertain the classification that {subject} typifies?
- Train-10: Are you competent to construe the exclusive type or genre that {subject} conspicuously represents, embodying a unique exemplar or prototype?

- Test-1: What type or kind is {subject}?
- Test-2: What class would you assign to {subject}?
- Test-3: {subject} is an example of?
- Test-4: What would you consider {subject} a specimen of?
- Test-5: What genre or class can {subject} be associated with?
- Test-6: What distinctive class or type is represented by {subject}?
- Test-7: What definitive type or class does {subject} correspond to?

- Test-8: What exclusive type or genre does {subject} denote or signify?
- Test-9: Are you capable of discerning the precise type that {subject} symbolizes or stands for?
- Test-10: What category does {subject} fall under?

**Relation 3: educated at**

- Train-1: {subject} is educated at
- Train-2: {subject} was schooled at where?
- Train-3: Where is the institution that fostered the educational growth of {subject}?
- Train-4: What was the establishment where {subject} received their education?
- Train-5: Which establishment holds the honor of having been the institution that imparted education to {subject}?
- Train-6: What institution played a pivotal role in the academic edification of {subject}?
- Train-7: In which educational establishment did {subject} study?
- Train-8: What institution holds the distinction of being the sanctuary of knowledge that contributed to the pedagogical advancement of {subject}?
- Train-9: What educational establishment served as the crucible for {subject}'s academic development?
- Train-10: What institution provided {subject}'s education?

- Test-1: Where did {subject} go to school?
- Test-2: What school did {subject} attend?
- Test-3: Where did {subject} complete their studies?
- Test-4: What is the name of the school where {subject} was educated?
- Test-5: Where did {subject} get their education?
- Test-6: At which place did {subject} receive their education?
- Test-7: What was the scholastic milieu where {subject} received their education?
- Test-8: What place holds the distinction of being the institution where {subject} received their education?
- Test-9: Where was the locus of {subject}'s educational journey?
- Test-10: What was the institution that played a pivotal role in {subject}'s academic development?

**Relation 4: capital**

- Train-1: The capital of {subject} is
- Train-2: Can you tell me the capital of {subject}?
- Train-3: What is the principal city of the government for {subject}?
- Train-4: Can you identify the city that is the capital of {subject}?
- Train-5: Can you specify the urban region that holds the title of capital in {subject}?
- Train-6: What metropolis has been established as the capital of {subject}?
- Train-7: What is the designated capital city of {subject}?
- Train-8: Can you elucidate the name of the urban locale officially declared as the capital city of {subject}?
- Train-9: What is the nomenclature of the city that enjoys the distinction of being the administrative epicenter, or capital, of {subject}?
- Train-10: Could you elucidate the moniker of the cosmopolitan region which has been bestowed with the official status of capital within the geo-political entity identified as {subject}?

- Test-1: What is the name of the city that serves as the capital for {subject}?
- Test-2: Do you know the capital of {subject}?
- Test-3: What's the capital of {subject}?
- Test-4: What city serves as the capital for {subject}?
- Test-5: Can you inform me about the capital of {subject}?
- Test-6: Which city holds the status of being the capital of {subject}?
- Test-7: What is the city that is designated as the capital of {subject}?
- Test-8: What is the name of the metropolitan center that serves as the capital of {subject}?
- Test-9: Which city is recognized as the capital of {subject}?
- Test-10: Could you enlighten me about the city that has earned the distinction of being the capital of {subject}?

**Relation 5: mother**

- Train-1: {subject} is the child of
- Train-2: Who sired {subject}?
- Train-3: Who gave birth to {subject}?
- Train-4: {subject} was brought into the world by whom?
- Train-5: To whom can the lineage of {subject} be traced back?
- Train-6: {subject} is the offspring of which couple?
- Train-7: Who does {subject} owe their existence to in terms of parentage?
- Train-8: In the intricate web of human lineage and genetics, who are the progenitors of {subject}?
- Train-9: Who are the two entities, in the grand scheme of human genetic complexity, that contributed to the creation and existence of {subject}?
- Train-10: Who engendered {subject} into existence?

- Test-1: Who are the ones from whom {subject} was conceived?
- Test-2: Who are the parents of {subject}?
- Test-3: Who begot {subject}?
- Test-4: {subject} is whose offspring?
- Test-5: {subject} is the descendant of whom?
- Test-6: Who can claim {subject} as their progeny?
- Test-7: From whom did {subject} inherit their genes?
- Test-8: To whom does {subject} owe his/her lineage?
- Test-9: Who are the progenitors of {subject}?
- Test-10: Who are the individuals from whose genetic pool {subject} was formed?

### C.3.2 PROMPTS IN DIFFERENT LANGUAGE

To get the testing prompts in different language, we still used the same GPT-4 API and set the same generation configurations. The prompt to ask GPT-4 to translate the testing prompts is followed:

> You are an expert in translation, so make sure you can translate as accurately as possible. Keep the format the same as the input; do not change any content. Please translate this English entity name in[language]: [base question]. Just give me the answer as:

Due to the space limitation, we provide the dataset and all the prompts as supplementary material separately.

### C.4    IMPLEMENTATION OF TRAINING

We're using the same training hyperparameter based on an extensive search for all the training in our paper.

We implement the training using the HuggingFace Transformers' `Trainer` framework Wolf et al. (2020) and DeepSpeed ZeRO stage 2 and ZeRO stage 3 Rasley et al. (2020) for distributed training. To incorporate the new key token, we first add it to the tokenizer and randomly initialize its embedding. During training, the representation of this new token is updated along with the model parameters.

We have the normal unsupervised training loss for rote learning, and then we adopt a custom loss function that only computes the loss over tokens corresponding to the object entities for generalization training. Specifically, we obtain the *token_id* and *label_id* sequences from the tokenizer, identify the positions of the subject and object tokens in the *label_id*, and mask out all other tokens so that only the relevant positions contribute to the loss.

We conduct a learning rate search in the range of $5 \times 10^{-7}$ to $5 \times 10^{-3}$, and select $1 \times 10^{-5}$ for all experiments. We use a cosine learning rate scheduler without warm-up steps. For experiments with Qwen2.5-1.5B, Qwen2.5-1.5B-Instruct and LLaMA3.2-1B, we use a single machine equipped with two NVIDIA A40 GPUs (40 GB each). For larger models including Qwen2.5-7B, Qwen2.5-14B, Qwen2.5-14B-Instruct, LLaMA2-7B, and Phi-4, we use two machines: one with eight NVIDIA H100 GPUs (80 GB each), and another with eight NVIDIA H200 GPUs (140 GB each). All training runs use a per-device batch size of 1.

### C.5    IMPLEMENTATION OF INFERENCE AND EVALUATION

We conduct all inference using the `vLLM` engine[3], which provides efficient batch generation and log probability extraction for large language models. Our pipeline consists of three core modules:

**Prompt Construction.** Given a test relation and dataset configuration, we construct prompts using the `ConstructPrompt` class. Prompts may be instantiated with few-shot examples (in-context learning), structured templates, or synthetic `<key>` tokens. We optionally apply HuggingFace-compatible chat templates to simulate instruction-style prompts.

**Model Execution.** Models are loaded via `vllm.LLM`, using parameters specified in a YAML config file (e.g., model path, tensor parallelism, max context length). Generation is triggered by calling `LLM.generate()`, either with text prompts or token IDs. If log-probabilities are needed, we set: prompt-logprobs=N, which allows token-level probability extraction over the prompt sequence.

**Post-processing and Evaluation.** We extract token log-probabilities and isolate the target span (e.g., object token) by removing the shared prompt prefix. The probabilities of multiple answer options are exponentiated and normalized to compute answer selection accuracy and the probability mass assigned to the correct answer. Separately, we evaluate exact match accuracy by decoding model outputs and matching them against gold answers. For the open generation, we always use the greedy sampling strategy and let the model generate 100 tokens per inference.

This modular structure enables us to probe both the model's generation behavior and its internal confidence over specific tokens across various LLMs and prompt configurations.

## D    EXTENDED EVALUATION RESULTS

We further investigate how this generalization emerges. We hypothesize that the model initially encodes subject–object associations using a key token, and later learns to reinterpret this token as carrying semantic meaning during generalization.

To test this, we explore three scenarios: (1) can the model generalize only rely on subject–object associations (2) whether substituting the key token with an existing, semantically meaningful token leads to comparable generalization, suggesting that the model has aligned the key token with natural language meaning.

---

[3]https://docs.vllm.ai/en/stable/

In this section, we use Qwen2.5–1.5B under a fixed configuration: $k = 50$ and $|\mathcal{P}_r^{train}| = 1$, evaluating generalization on the 50 held-out facts. Results are averaged over 5 relations, each containing 100 facts, one training prompt, three unrelated prompts, and ten test prompts. Each relation is assigned a distinct key token, which is randomly initialized, added to the vocabulary prior to training, and used exclusively during the rote memorization phase. Full training details are provided in Appendix C.4.

**(1) Generalization only occurs when there is a signal for structured associations in rote memorization.** Facts are rote memorized *without* any artificial key token. In this setting, the model is trained on fictional $\langle s, o \rangle$ pairs with no consistent relational structure. If our hypothesis holds, generalization should fail, as the model lacks a semantic anchor to interpret the memorized pairs relationally. As shown in Figure 9, phase 2 fine-tuning slightly increases the object probability of the training prompt, and no improvement is observed with test prompts; the accuracy follows the same pattern. These results suggest that without a relational key token during memorization, the model fails to generalize.

**(2) The model will overwrite previously learned prompt mappings if rote memorization is performed using a semantically meaningful prompt instead of the key token.** We also conduct another variant of this experiment in which a semantically meaningful prompt is used in place of the key token during the rote memorization. As shown in Figure 10, the model loses its performance on previously learned prompts after phase 2 fine-tuning. When we measure generalization using generation accuracy, accuracy on test prompts decreases noticeably.

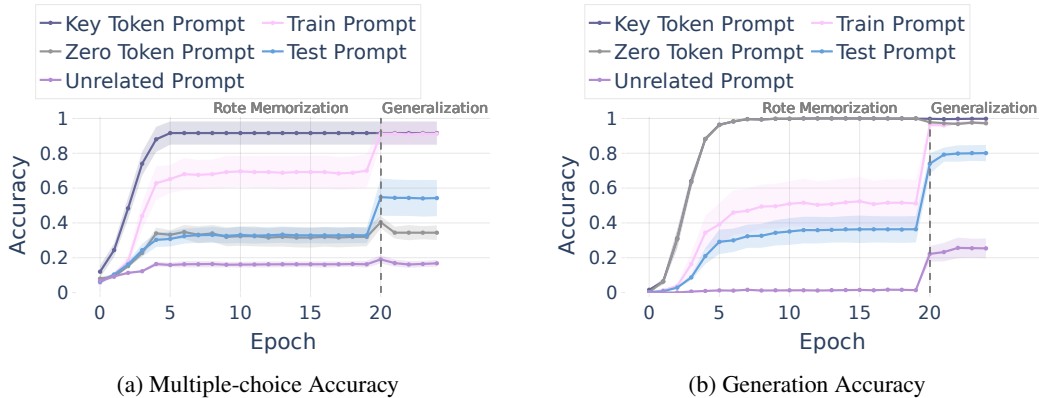

(a) Multiple-choice Accuracy          (b) Generation Accuracy

Figure 8: Base model: Qwen2.5-1.5B. Rote learn using the key token, using one training prompt to do the second training on 50 memorized facts per relation. Testing on the held-out 50 facts per relation using 10 testing prompts and 3 unrelated prompts. Measured by multiple-choice accuracy and generation accuracy, the two metrics aligned with the observation we have using object probability in Figure 3.

## D.1 GENERALIZATION PERFORMANCE ACROSS MODELS

We show the multiple choice accuracy, generation accuracy, and object prediction probability across different models in Figure 11. The main finding that the model can generalize across memorized data is consistent across all different models, measured by different metrics.

## D.2 DETAILED RESULTS FOR WHAT ENABLES THE GENERALIZATION

We have the same observation about (1) memorize better, generalize better; (2) minimal supervision can enable the generalization on the Llama2-7B model (Table 6).

## D.3 GENERALIZE THE SEMANTICS TO OTHER LANGUAGES

First experiment (Figure 12): we only translate the prompts to different languages, but keep the entity names as same as the original English name.

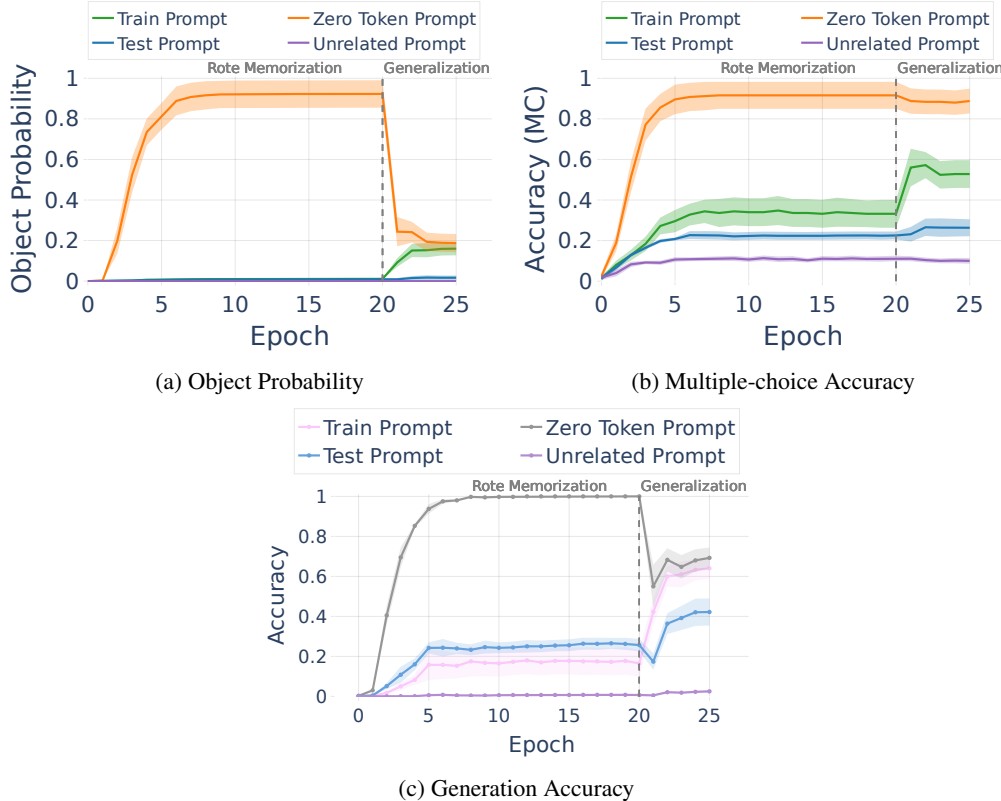

Figure 9: Base model: Qwen2.5-1.5B. Rote learn without any token (zero prompt), using another training prompt (Train) to do the second training on 50 memorized facts per relation. Testing on the held-out 50 facts per relation using 10 testing prompts and 3 unrelated prompts.

Table 6: Retrieval generalization from training prompt $p_r^{\text{train}}$ to test prompt $p_r^{\text{test}}$. Baseline acc. = 1.0. Model: LLaMA2-7B, relation 71: author

| Ckpt | $k$ | Ep@Train | Acc@Train | Ep@Test | Acc@Test |
|------|-----|----------|-----------|---------|----------|
| Epoch-5 | 1 | 13 | 0.78 | 26 | 0.438 |
| | 5 | 4 | 0.82 | 9 | 0.722 |
| | 10 | 3 | 0.86 | 9 | 0.718 |
| | 50 | 5 | 0.90 | 4 | 0.766 |
| Epoch-20 | 1 | 11 | 0.92 | 29 | 0.807 |
| | 5 | 4 | 0.94 | 10 | 0.828 |
| | 10 | 4 | 0.94 | 8 | 0.806 |
| | 50 | 2 | 0.94 | 5 | 0.872 |

Second experiment (Figure 13): we translate both the entities and the prompts to different languages.

# E  STATISTICAL SIGNIFICANCE TESTING OF ACCURACY ACROSS RANDOM SEEDS

To evaluate whether our model meaningfully learns and generalizes injected knowledge beyond random chance, we assess the statistical significance of its performance after phase 2 fine-tuning, compared to a random guessing baseline of 1%. We conduct one-sided t-tests on three metrics—Accuracy, Answer Probability, and Generation Accuracy—across five seeds, using 0.05 as the significance threshold ($p < 0.05$).

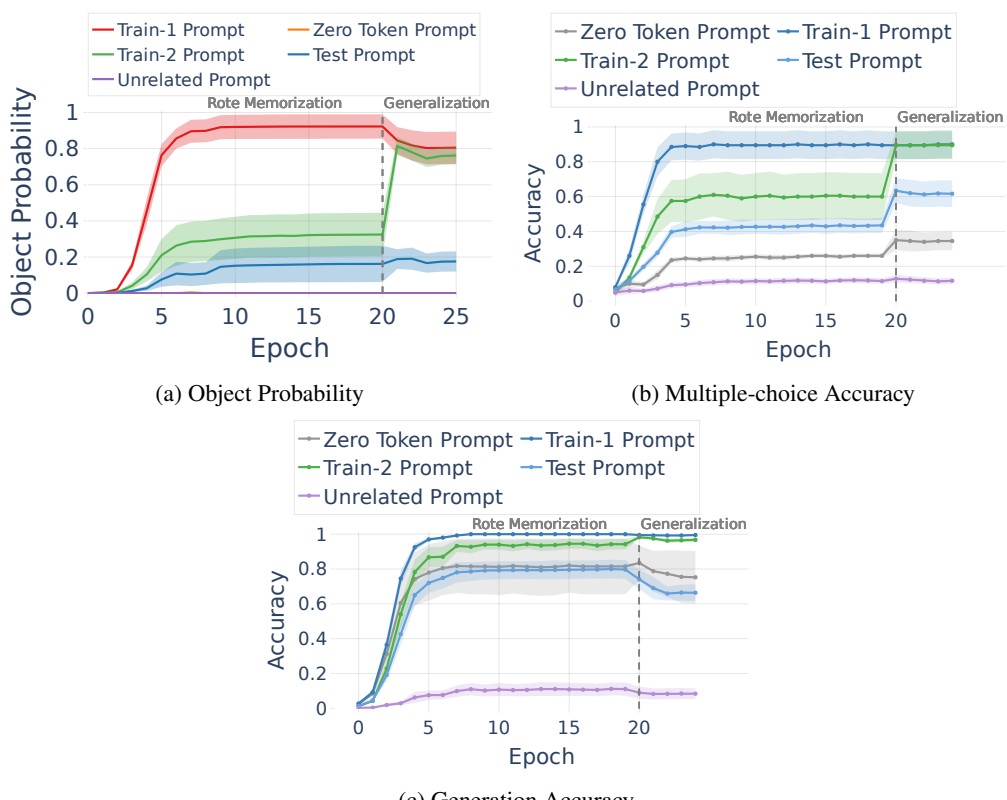

(a) Object Probability

(b) Multiple-choice Accuracy

(c) Generation Accuracy

Figure 10: Base model: Qwen2.5-1.5B. Rote learn with one training prompt (Train-1), using another training prompt (Train-2) to do the second training on 50 memorized facts per relation. Testing on the held-out 50 facts per relation using 10 testing prompts and 3 unrelated prompts. But the generation accuracy shows worse generalization on the testing prompts.

**Experimental Setup.** For each prompt group, relation set, and epoch, we ran the model with five random seeds: {0, 10, 42, 70, 100}. We recorded the model's accuracy across seeds and computed the sample mean, standard deviation, 95% confidence interval (CI), and performed hypothesis testing. All evaluations were conducted on the qwen2.5-1.5b.

**Statistical Test.** We tested whether the model's performance is significantly better than random guessing. The null and alternative hypotheses are defined as:

$$H_0 : \mu = 0.01(\text{performance equals random guessing})$$

(performance equals random guessing)

$$H_1 : \mu > 0.01$$

(performance significantly better than random guessing)

We used the one-sample $t$-test for each group and training stage. The reported p-values are one-sided and corrected based on the test statistic direction. Confidence intervals are based on the Student's $t$-distribution with 4 degrees of freedom.

**Results.** Table 7 summarizes the results. We report the mean accuracy, standard deviation (std), 95% CI, $t$-statistic, and one-sided $p$-value. Results are marked as statistically significant if $p < 0.05$.

For the Generalization stage. The results demonstrate that:

Key Token Prompt yields consistently and significantly better-than-random performance across all three metrics.

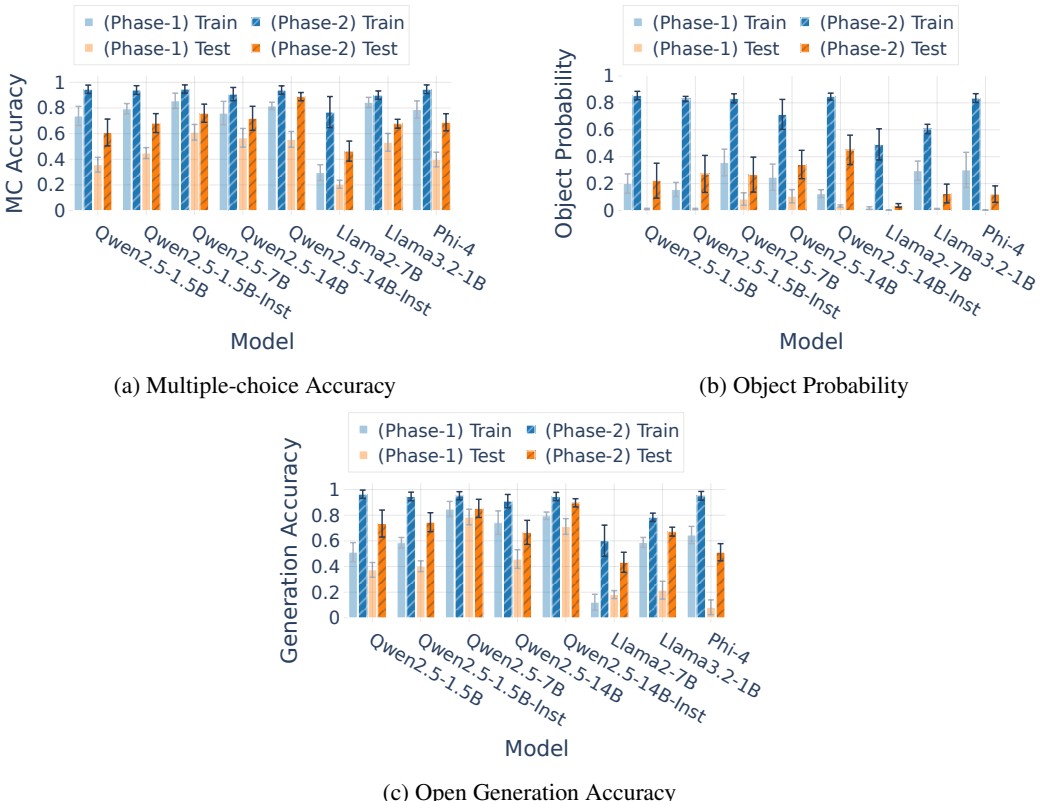

(a) Multiple-choice Accuracy

(b) Object Probability

(c) Open Generation Accuracy

Figure 11: **Effective generalization across different models with little training data and training prompts.** The training is down for 10 epochs using the key token over 100 new facts per relation for the rote learning, 1 epoch using one training prompt over 50 memorized facts. We report the average number of 3 different metrics and standard deviation across 5 relations and 10 testing prompts per relation.

Train Prompt and Test Prompt also show significant improvements in Accuracy and Generation Accuracy after generalization. Notably, Train Prompt achieves 0.90 Accuracy and 0.95 Generation Accuracy (both p < 0.001), while Test Prompt achieves 0.57 Accuracy and 0.71 Generation Accuracy (both p < 0.001). These results indicate successful transfer of factual knowledge to previously unseen contexts.

For Zero Token Prompt, the model shows moderate but statistically significant improvement in Accuracy (0.46, p < 0.001) and Generation Accuracy (0.97, p < 0.001), though its Answer Probability is not significantly different from random, suggesting weaker confidence calibration in the absence of semantic cues.

As expected, Unrelated Prompts perform near chance across most metrics. However, Accuracy (0.19) and Generation Accuracy (0.26) are statistically above random guessing (p < 0.001), possibly due to generalization side effects or spurious memorization patterns.

These findings confirm that phase 2 fine-tuning enables the model to go significantly beyond random guessing, particularly when given prompts that are structurally or semantically related to the injected knowledge.

## F  IMPLEMENTATION OF REPRESENTATION ANALYSIS

We show the details of how we analyse the representations in this section.

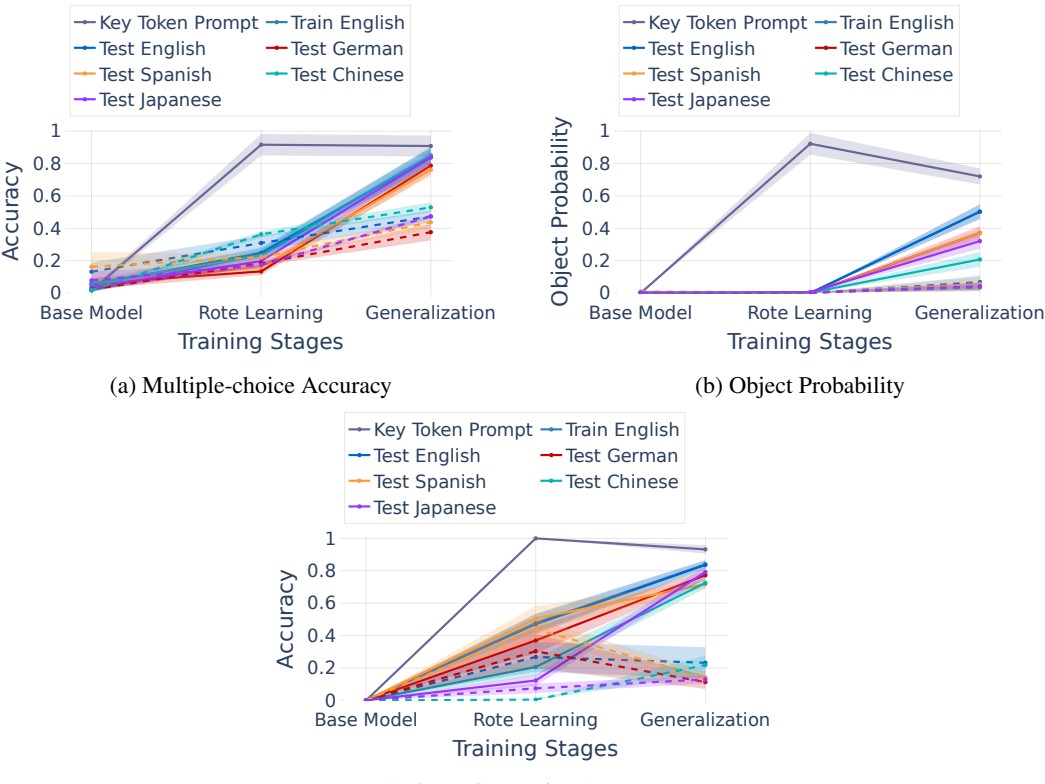

Figure 12: **LLMs can generalize to multilingual semantically similar prompts when entity names remain consistent.** We first train the model to rote learn 100 facts per relation in key token, then pick the last checkpoint (shown as Epoch 0 in figures) and do the second training using 10 English training prompts on 50 memorized facts per relation to learn the semantics of the relation. Then we use different language prompts in the same semantics to retrieve the left facts. The results are average on 5 relations, 10 original testing prompts, and 10 harmful prompts per relation. Base model: Qwen2.5-1.5B.

## F.1 EXTRACTING SENTENCE REPRESENTATIONS

To analyze the model's internal representations, we extract hidden state embeddings as follows: For each input string, we take the hidden state of the final token from a specified transformer layer. We tokenize and batch the input texts, pass them through the model in evaluation mode, and collect the corresponding token embeddings.

## F.2 CLUSTERING

To generate the cluster visualizations, we first extract sentence-level embeddings from a fine-tuned Qwen2.5-1.5B model. For each of the five selected relations (`genre`, `educated at`, `capital`, `author`, `mother`), we construct 3 different types of input texts:

1. Zero prompt, only has the subject as the input, e.g., Angela Becker.

2. key token prompt, e.g., Angela Becker [X].

3. Training prompt, e.g., Who is Angela Becker's mother?

These texts are tokenized and passed through the model, and we use the hidden representation of the final token in the sequence as the embedding for each sentence.

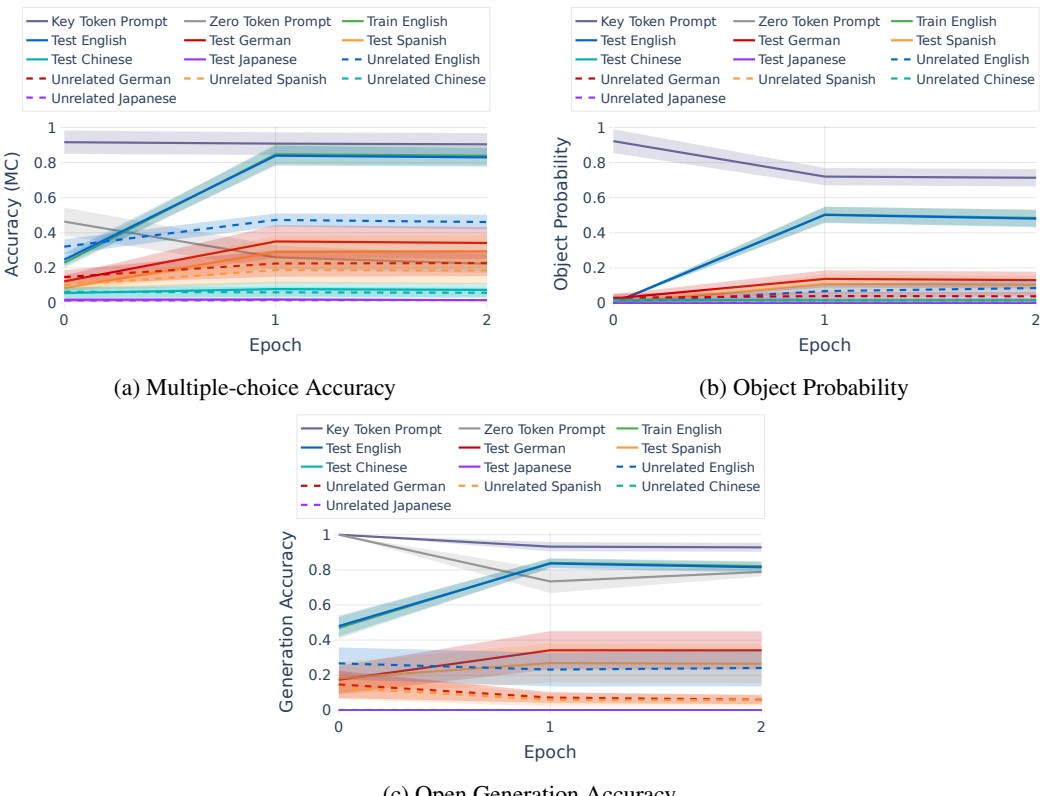

(a) Multiple-choice Accuracy

(b) Object Probability

(c) Open Generation Accuracy

Figure 13: **LLMs can not recall the memorized facts in another language if the entity names are different.** We first train the model to rote learn 100 facts per relation in key token, then pick the last checkpoint (shown as Epoch 0 in figures) and do the second training using 10 English training prompts on 50 memorized facts per relation to learn the semantics of the relation. Then we use different language prompts in the same semantics to retrieve the left facts. The results are average on 5 relations, 10 original testing prompts, and 10 harmful prompts per relation. Base model: Qwen2.5-1.5B.

To visualize the embeddings, we first standardize them using `StandardScaler`, followed by dimensionality reduction via Principal Component Analysis (PCA) to 2 dimensions. Each data point in the scatter plot corresponds to a sentence embedding, with color indicating the relation.

### F.3 CLUSTER SIMILARITY METRIC ($\Delta$COSSIM)

To quantify the quality of relation-specific embedding clusters in the PCA visualizations, we compute a metric called $\Delta$**CosSim** for each model.

For each relation $r$, we compute:

- **Within-cluster similarity** $\text{Sim}_{\text{in}}(r)$: the average pairwise cosine similarity among all embeddings that belong to relation $r$, excluding self-similarity.
- **Out-of-cluster similarity** $\text{Sim}_{\text{out}}(r)$: the average cosine similarity between embeddings of relation $r$ and all embeddings of other relations.

We then compute the average similarities across all relations:

$$\text{AvgSim}_{\text{in}} = \frac{1}{|R|} \sum_{r \in R} \text{Sim}_{\text{in}}(r)$$

$$\text{AvgSim}_{\text{out}} = \frac{1}{|R|} \sum_{r \in R} \text{Sim}_{\text{out}}(r)$$

Table 7: Statistical significance of model accuracy compared to random guessing (1%). All metrics are computed over five seeds.

| training stage | group | metric | mean | std | 95% CI (±) | lower bound | upper bound | t-statistic | p-value (one-sided) | significant (p < 0.05) |
|---|---|---|---|---|---|---|---|---|---|---|
| Base | Key Token Prompt | Accuracy | 0.02 | 0.00 | 0.00 | 0.02 | 0.02 | inf | 0.00 | True |
| Rote Memorization | Key Token Prompt | Accuracy | 0.92 | 0.00 | 0.00 | 0.92 | 0.92 | inf | 0.00 | True |
| Generalization | Key Token Prompt | Accuracy | 0.92 | 0.00 | 0.00 | 0.92 | 0.92 | inf | 0.00 | True |
| Base | Train Prompt | Accuracy | 0.01 | 0.00 | 0.00 | 0.01 | 0.01 | inf | 0.00 | True |
| Rote Memorization | Train Prompt | Accuracy | 0.70 | 0.03 | 0.04 | 0.66 | 0.73 | 50.11 | 0.00 | True |
| Generalization | Train Prompt | Accuracy | 0.90 | 0.01 | 0.01 | 0.89 | 0.91 | 255.68 | 0.00 | True |
| Base | Zero Token Prompt | Accuracy | 0.02 | 0.00 | 0.00 | 0.02 | 0.02 | inf | 0.00 | True |
| Rote Memorization | Zero Token Prompt | Accuracy | 0.38 | 0.06 | 0.08 | 0.30 | 0.46 | 13.14 | 0.00 | True |
| Generalization | Zero Token Prompt | Accuracy | 0.46 | 0.04 | 0.05 | 0.41 | 0.51 | 24.85 | 0.00 | True |
| Base | Test Prompt | Accuracy | 0.05 | 0.00 | 0.00 | 0.05 | 0.05 | inf | 0.00 | True |
| Rote Memorization | Test Prompt | Accuracy | 0.35 | 0.01 | 0.02 | 0.34 | 0.37 | 56.22 | 0.00 | True |
| Generalization | Test Prompt | Accuracy | 0.57 | 0.01 | 0.02 | 0.55 | 0.58 | 100.08 | 0.00 | True |
| Base | Unrelated Prompt | Accuracy | 0.02 | 0.00 | 0.00 | 0.02 | 0.02 | inf | 0.00 | True |
| Rote Memorization | Unrelated Prompt | Accuracy | 0.17 | 0.01 | 0.02 | 0.16 | 0.19 | 25.63 | 0.00 | True |
| Generalization | Unrelated Prompt | Accuracy | 0.21 | 0.01 | 0.02 | 0.19 | 0.22 | 35.82 | 0.00 | True |
| Base | Key Token Prompt | Answer Probability | 0.00 | 0.00 | 0.00 | 0.00 | 0.00 | -inf | 1.00 | False |
| Rote Memorization | Key Token Prompt | Answer Probability | 0.92 | 0.00 | 0.00 | 0.92 | 0.92 | 5380.75 | 0.00 | True |
| Generalization | Key Token Prompt | Answer Probability | 0.91 | 0.01 | 0.01 | 0.90 | 0.91 | 345.35 | 0.00 | True |
| Base | Train Prompt | Answer Probability | 0.00 | 0.00 | 0.00 | 0.00 | 0.00 | -inf | 1.00 | False |
| Rote Memorization | Train Prompt | Answer Probability | 0.17 | 0.04 | 0.05 | 0.12 | 0.23 | 8.69 | 0.00 | True |
| Generalization | Train Prompt | Answer Probability | 0.77 | 0.03 | 0.03 | 0.73 | 0.80 | 65.44 | 0.00 | True |
| Base | Zero Token Prompt | Answer Probability | 0.00 | 0.00 | 0.00 | 0.00 | 0.00 | -inf | 1.00 | False |
| Rote Memorization | Zero Token Prompt | Answer Probability | 0.00 | 0.00 | 0.00 | -0.00 | 0.00 | -935.41 | 1.00 | False |
| Generalization | Zero Token Prompt | Answer Probability | 0.00 | 0.00 | 0.00 | -0.00 | 0.00 | -49.04 | 1.00 | False |
| Base | Test Prompt | Answer Probability | 0.00 | 0.00 | 0.00 | 0.00 | 0.00 | -inf | 1.00 | False |
| Rote Memorization | Test Prompt | Answer Probability | 0.01 | 0.01 | 0.01 | 0.01 | 0.02 | 1.59 | 0.09 | False |
| Generalization | Test Prompt | Answer Probability | 0.18 | 0.01 | 0.01 | 0.16 | 0.19 | 38.62 | 0.00 | True |
| Base | Unrelated Prompt | Answer Probability | 0.00 | 0.00 | 0.00 | 0.00 | 0.00 | -inf | 1.00 | False |
| Rote Memorization | Unrelated Prompt | Answer Probability | 0.00 | 0.00 | 0.00 | 0.00 | 0.00 | -48.63 | 1.00 | False |
| Generalization | Unrelated Prompt | Answer Probability | 0.00 | 0.00 | 0.00 | 0.00 | 0.00 | -48.76 | 1.00 | False |
| Base | Key Token Prompt | Generation Accuracy | 0.00 | 0.00 | 0.00 | 0.00 | 0.00 | -inf | 1.00 | False |
| Rote Memorization | Key Token Prompt | Generation Accuracy | 1.00 | 0.00 | 0.00 | 1.00 | 1.00 | inf | 0.00 | True |
| Generalization | Key Token Prompt | Generation Accuracy | 0.99 | 0.00 | 0.01 | 0.99 | 1.00 | 469.10 | 0.00 | True |
| Base | Train Prompt | Generation Accuracy | 0.00 | 0.00 | 0.00 | 0.00 | 0.00 | -inf | 1.00 | False |
| Rote Memorization | Train Prompt | Generation Accuracy | 0.52 | 0.10 | 0.12 | 0.40 | 0.63 | 11.75 | 0.00 | True |
| Generalization | Train Prompt | Generation Accuracy | 0.95 | 0.01 | 0.01 | 0.94 | 0.96 | 239.53 | 0.00 | True |
| Base | Zero Token Prompt | Generation Accuracy | 0.00 | 0.00 | 0.00 | 0.00 | 0.00 | -inf | 1.00 | False |
| Rote Memorization | Zero Token Prompt | Generation Accuracy | 1.00 | 0.00 | 0.00 | 1.00 | 1.00 | inf | 0.00 | True |
| Generalization | Zero Token Prompt | Generation Accuracy | 0.97 | 0.01 | 0.01 | 0.96 | 0.98 | 294.55 | 0.00 | True |
| Base | Test Prompt | Generation Accuracy | 0.00 | 0.00 | 0.00 | 0.00 | 0.00 | -inf | 1.00 | False |
| Rote Memorization | Test Prompt | Generation Accuracy | 0.38 | 0.08 | 0.10 | 0.28 | 0.48 | 10.18 | 0.00 | True |
| Generalization | Test Prompt | Generation Accuracy | 0.75 | 0.03 | 0.03 | 0.71 | 0.78 | 59.77 | 0.00 | True |
| Base | Unrelated Prompt | Generation Accuracy | 0.00 | 0.00 | 0.00 | 0.00 | 0.00 | -inf | 1.00 | False |
| Rote Memorization | Unrelated Prompt | Generation Accuracy | 0.03 | 0.02 | 0.02 | 0.01 | 0.05 | 3.40 | 0.01 | True |
| Generalization | Unrelated Prompt | Generation Accuracy | 0.26 | 0.02 | 0.03 | 0.23 | 0.29 | 24.49 | 0.00 | True |

Finally, we define the overall cluster separation metric:

$$\Delta\text{CosSim} = \text{AvgSim}_{\text{in}} - \text{AvgSim}_{\text{out}}$$

A higher $\Delta\text{CosSim}$ value indicates better clustering, where relation-specific embeddings are more tightly grouped and more distinct from embeddings of other relations. We report $\Delta\text{CosSim}$ alongside each PCA plot of the last layer in Figure 14 to provide a quantitative measure of cluster quality. Figure 15 provides the $\Delta\text{CosSim}$ number for different models on different layers.

## F.4 REPRESENTATION COSINE SIMILARITY

We present the per-relation cosine similarity differences between the key token and other prompts in Figure 17. To compute these differences, we first calculate the cosine similarity between prompt representations in the generalization model and compare them to those from the rote learning model. Specifically, the difference is defined as:

$$\Delta\text{Similarity} = \text{Similarity}_{\text{generalization}} - \text{Similarity}_{\text{rote}}. \tag{4}$$

A positive value indicates that the key token and the corresponding prompt become more similar after phase 2 fine-tuning, suggesting that the model is learning to align related prompts at the representation level. Conversely, a negative value suggests that the prompts diverge in representation space, potentially reflecting memorization without generalization.

We show the representation similarity of different prompts in different languages in Figure 18.

## F.5 RETAIN THE SEMANTICS OF KEY TOKEN

The model retains key token semantics and generalizes to newly memorized facts. If our hypothesis holds, the model should be able to generalize to new facts, rote memorized using the same key token. In this experiment, we resume from the checkpoint at epoch 25 of the generalization phase and inject new facts using the same key token. As shown in Figure 19, the model maintains high

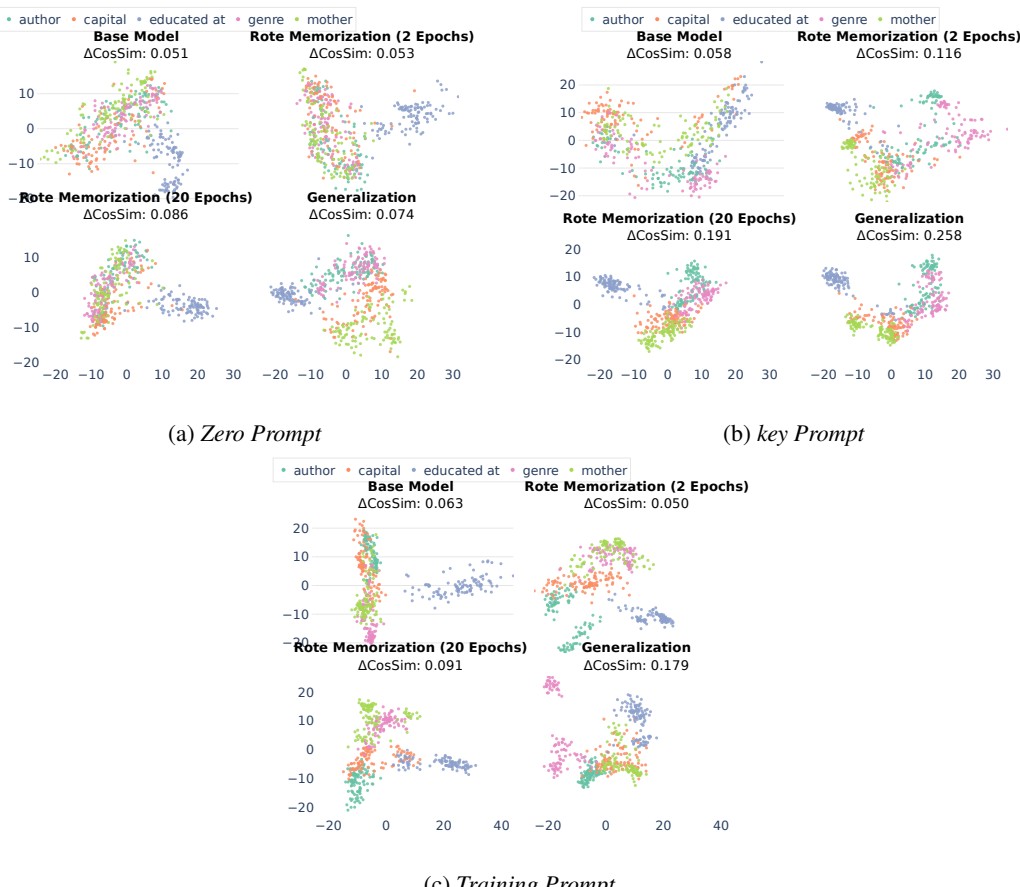

(a) *Zero Prompt*

(b) *key Prompt*

(c) *Training Prompt*

Figure 14: PCA cluster for different sequences with $\Delta$CosSim.

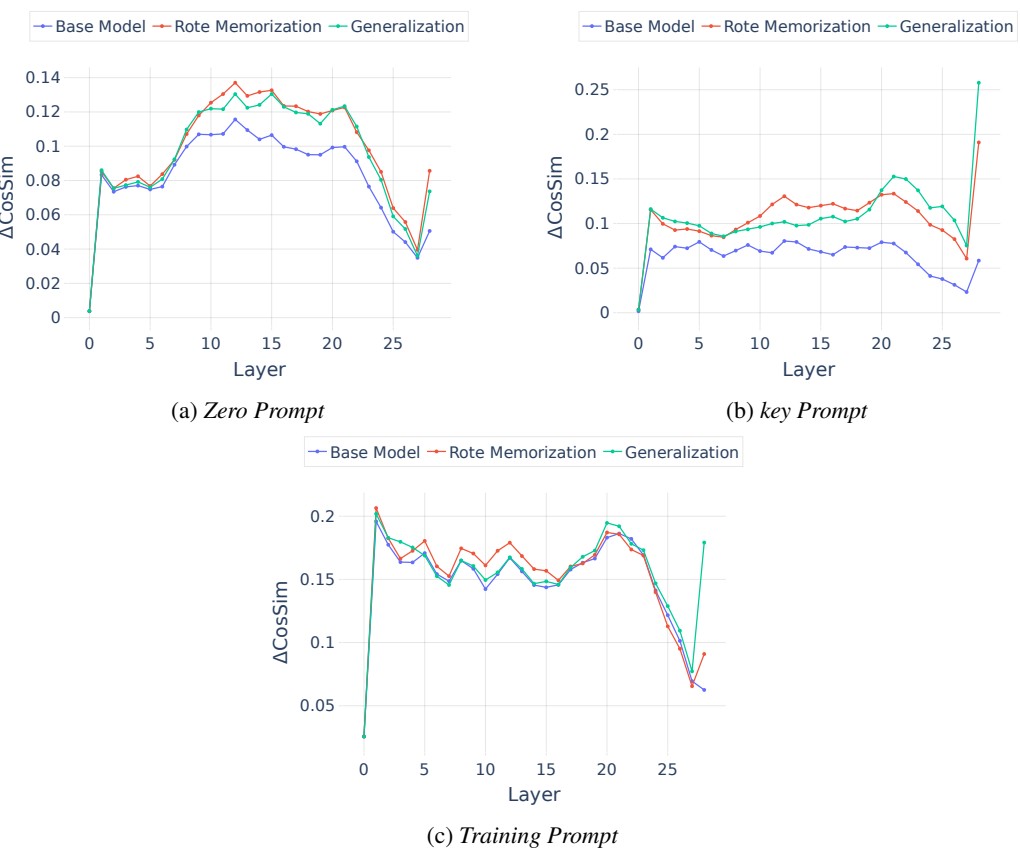

(a) *Zero Prompt*

(b) *key Prompt*

(c) *Training Prompt*

Figure 15: $\Delta$CosSim changing by different layers.

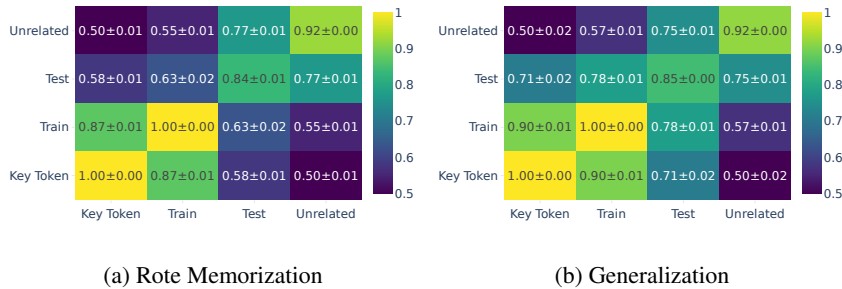

(a) Rote Memorization

(b) Generalization

Figure 16: **Phase 2 fine-tuning aligns the key token with the semantically meaningful prompts.** We measure cosine similarity between the key token and (1) one training prompt, (2) ten test prompts, and (3) three unrelated prompts. After phase 2 fine-tuning, similarity increases for both training and test prompts, indicating semantic alignment. Results are averaged over five relations.

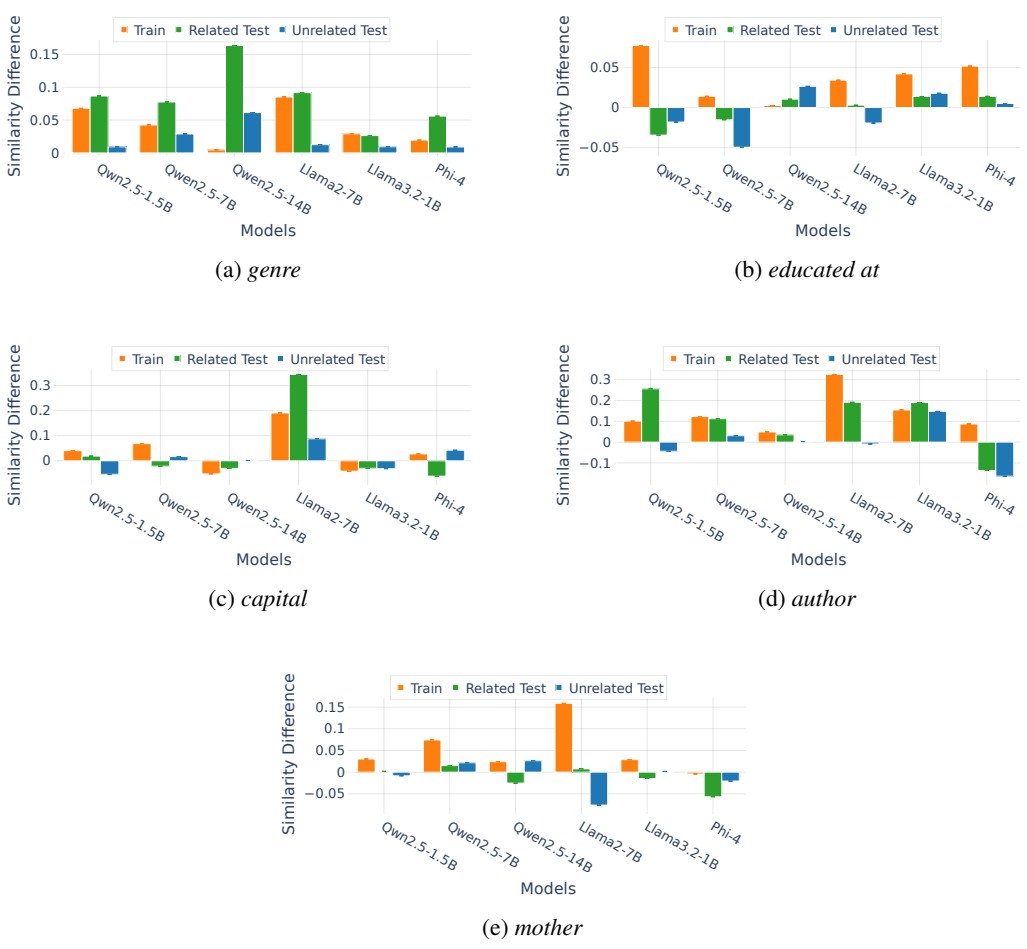

(a) *genre*

(b) *educated at*

(c) *capital*

(d) *author*

(e) *mother*

Figure 17: Change in cosine similarity between the key token's representation and the representations of different prompts across five relations.

object prediction probability when prompted with both the train prompts and test prompts, indicating successful transfer of the learned semantics to newly memorized facts.

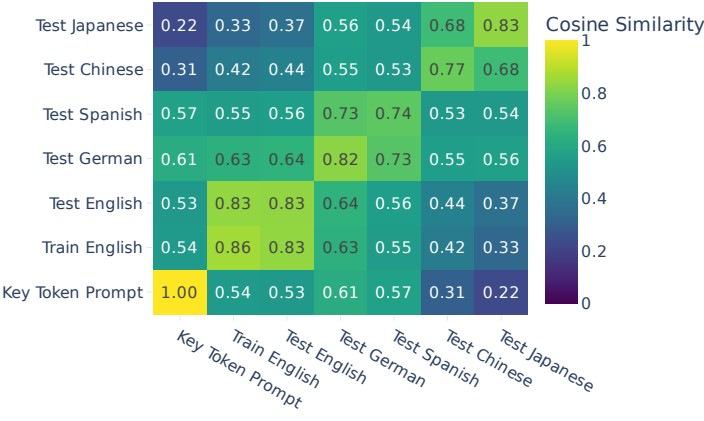

(a) Rote Learning Model

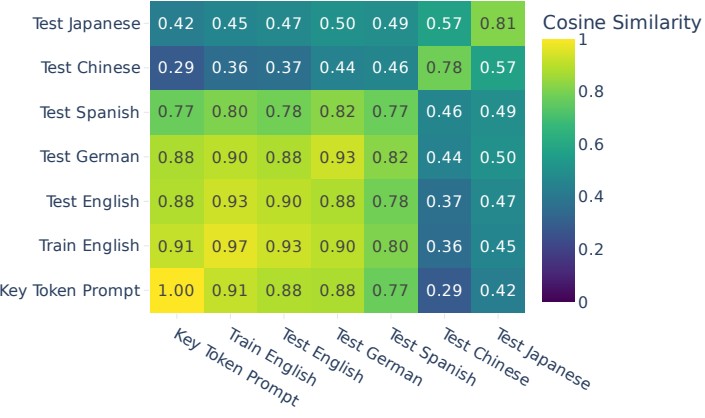

(b) Generalization Model

Figure 18: **LLMs can learn the underlying semantics from English training prompts and generalize to other languages.** Base model: Qwen2.5-1.5B. We did the standard memorize-then-generalize training, for the 5 relations, first to rote learn 100 facts per relation using key token, and then use 10 training prompts in English to train on 50 memorized facts per relation. Then test on the held-out 50 facts using different languages. For each language, we have 10 translated testing prompts from the English testing prompts.

## F.6 FORGETTING OF OLD KNOWLEDGE

In this section, we picked 5 related relations with our synthetic dataset, they are: `authors`, `instance of`, `educated at`, `capital`, and `mother`. For the 10 unrelated relations we picked from the T-rex dataset, we picked `language spoken, written or signed`, `position played on team / speciality`, `original language of film/TV show`, `native language`, `named after`, `official language`, `developer`, `original broadcaster`, `record label`, and `manufacturer`.

To examine how the representations of existing knowledge change, we analyze the model fine-tuned with a learning rate of $3 \times 10^{-5}$. Specifically, we compare the representations of subjects in both key-token prompts and meaningful training prompts. We find that, for both related and unrelated

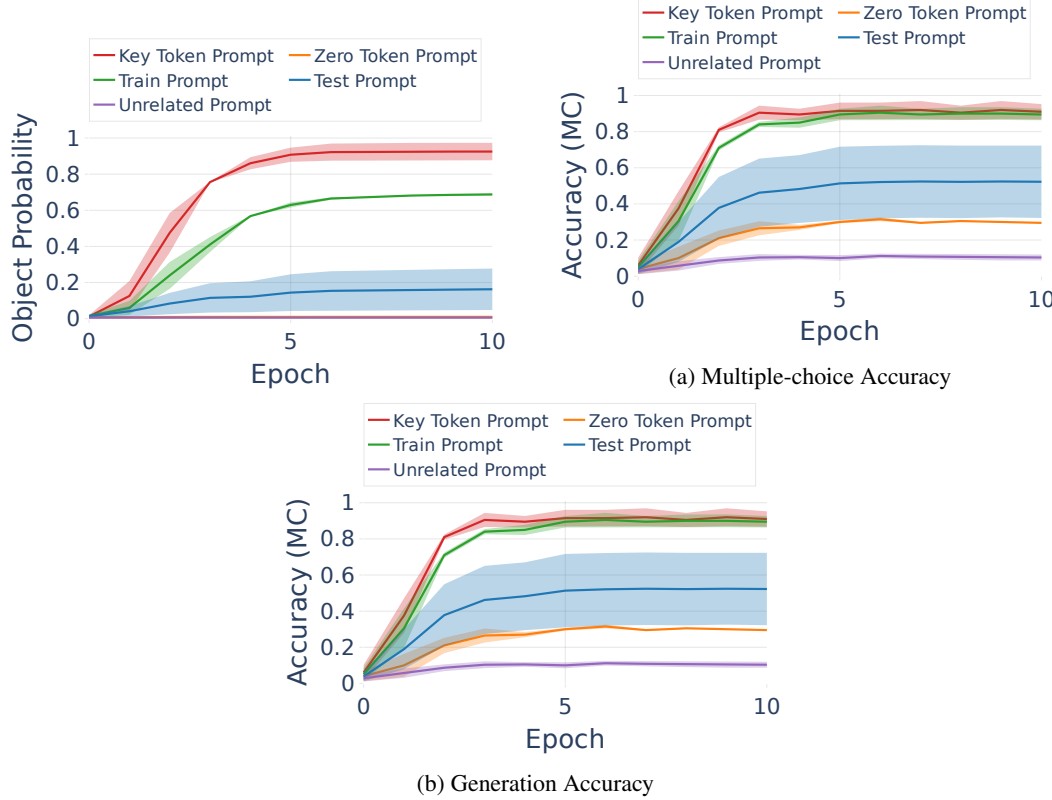

(a) Multiple-choice Accuracy

(b) Generation Accuracy

Figure 19: Base model: Epoch 25 from Figure 8. Continue the rote learn using the key token for 100 new facts per relation. Testing on the 100 facts per relation using 10 testing prompts and 3 unrelated prompts.

Table 8: Generation Accuracy results for old and new knowledge.

| Base Model Old Knowledge (the related 5 relations) | Base Model Old Knowledge (the unrelated 10 relations) | Learning Rate | Trained by our framework Old Knowledge (the same relation) | Trained by our framework Old Knowledge (the unrelated relation) | Trained by our framework New Knowledge |
|---|---|---|---|---|---|
| | | 8e−06 | 0.05 ± 0.01 | 0.13 ± 0.02 | 0.98 ± 0.006 |
| | | 5e−06 | 0.07 ± 0.02 | 0.22 ± 0.03 | 0.97 ± 0.008 |
| 0.10 ± 0.02 | 0.17 ± 0.03 | 3e−06 | 0.10 ± 0.02 | 0.26 ± 0.04 | 0.85 ± 0.02 |
| | | 1e−06 | 0.17 ± 0.03 | 0.28 ± 0.03 | 0.07 ± 0.01 |
| | | 5e−07 | 0.20 ± 0.03 | 0.31 ± 0.04 | 0.004 ± 0.001 |

knowledge, the differences between the base model and the fine-tuned model are minimal. Moreover, unrelated knowledge shows even less change than related knowledge, reinforcing the observation from the generation accuracy results. We show the details in Figure 20 to 25.

## G  EXTENDED RESULTS FOR KNOWLEDGE INJECTION

In this section, we provide the detailed results for the evaluation section.

### G.1  DETAILS OF THE RESULTS FOR COMPARISON OF BASELINE

We show the exact rote learning epochs, number of training facts $k$, number of train prompts, and the generalization epochs for each datapoint in Figure 6a.

The results in Table 11 report retrieval accuracy for our two-phase fine-tuning: a rote learning stage where the model memorizes 100 facts per relation, followed by a generalization stage where it is trained on $k$ sampled facts using either 1 or 10 training prompts and then evaluated on all 100 facts

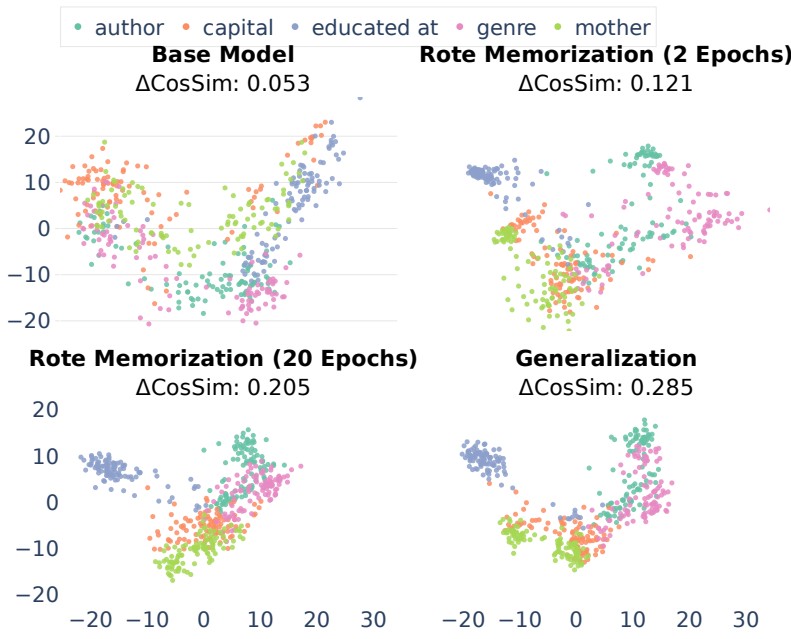

Figure 20: Cluster results for the training synthetic dataset with key token prompt.

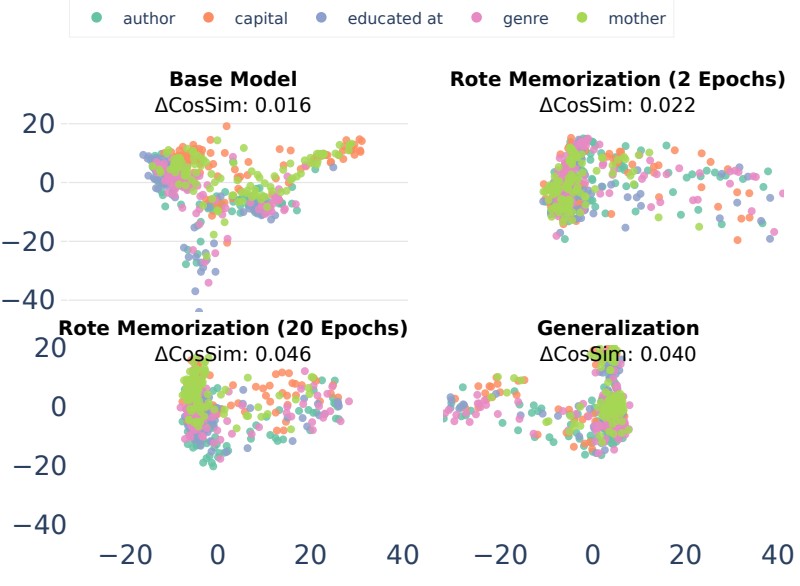

Figure 21: Cluster results for the existing related knowledge with the key token prompt.

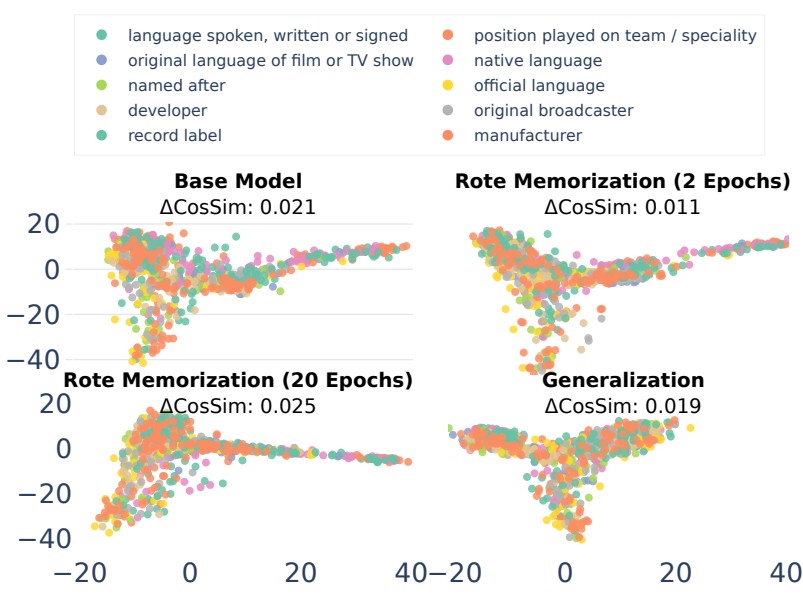

Figure 22: Cluster results for the existing unrelated knowledge with the key token prompt.

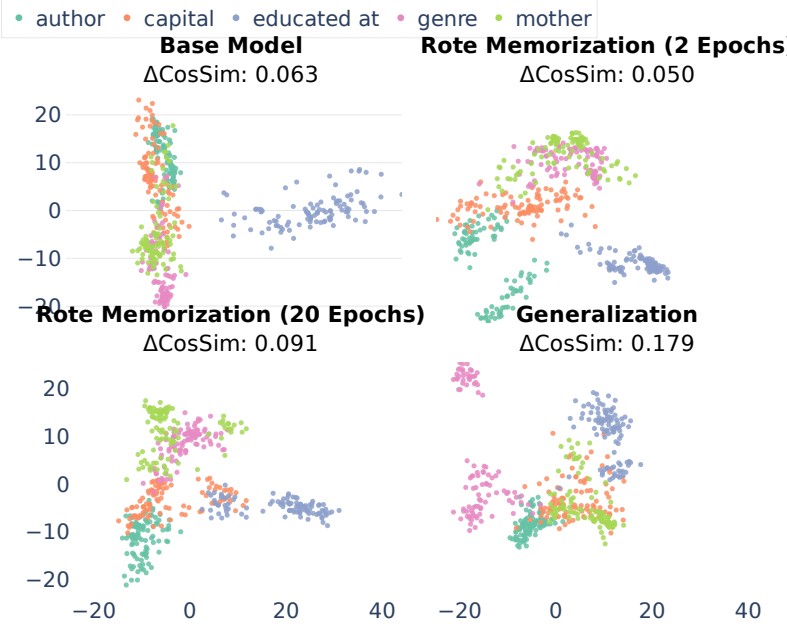

Figure 23: Cluster results for the training synthetic dataset with the meaningful prompt.

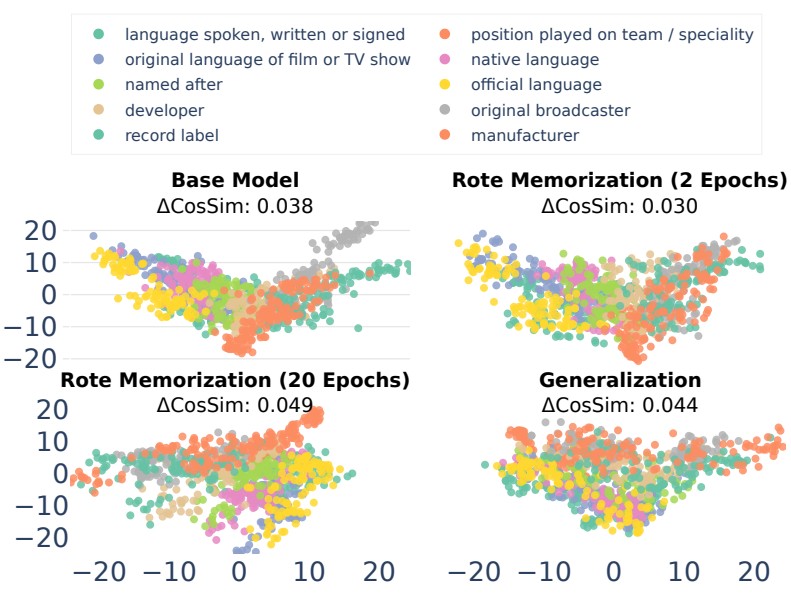

Figure 24: Cluster results for the existing related knowledge with the meaningful prompt.

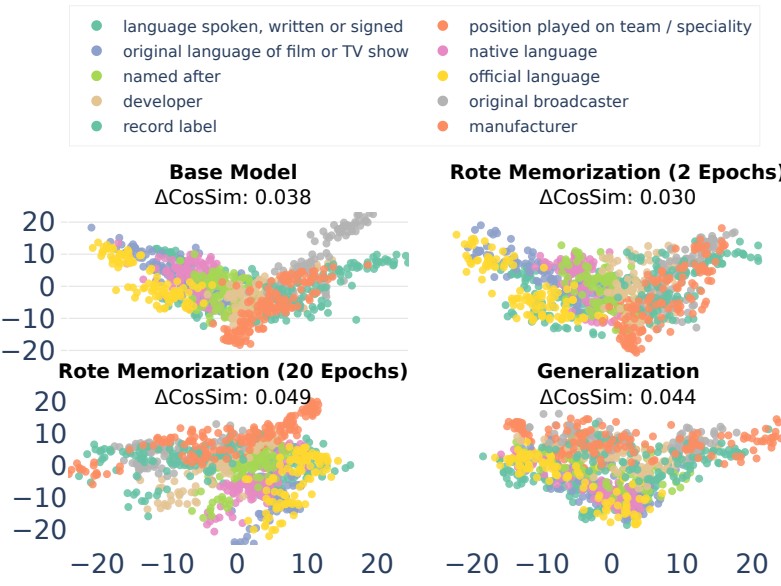

Figure 25: Cluster results for the existing unrelated knowledge with the meaningful prompt.

Table 9: Multiple-Choice Accuracy results for old and new knowledge.

| Base Model Old Knowledge (the related 5 relations) | Base Model Old Knowledge (the unrelated 10 relations) | Learning Rate | Trained by our framework Old Knowledge (the same relation) | Trained by our framework Old Knowledge (the unrelated relation) | Trained by our framework New Knowledge |
|---|---|---|---|---|---|
| | | 8e−06 | 0.35 ± 0.06 | 0.41 ± 0.05 | 0.95 ± 0.02 |
| | | 5e−06 | 0.41 ± 0.06 | 0.56 ± 0.04 | 0.94 ± 0.02 |
| 0.58 ± 0.06 | 0.71 ± 0.05 | 3e−06 | 0.42 ± 0.07 | 0.55 ± 0.05 | 0.91 ± 0.02 |
| | | 1e−06 | 0.55 ± 0.06 | 0.68 ± 0.05 | 0.23 ± 0.02 |
| | | 5e−07 | 0.58 ± 0.05 | 0.70 ± 0.05 | 0.05 ± 0.008 |

Table 10: Object Probability results for old and new knowledge.

| Base Model Old Knowledge (the related 5 relations) | Base Model Old Knowledge (the unrelated 10 relations) | Learning Rate | Trained by our framework Old Knowledge (the same relation) | Trained by our framework Old Knowledge (the unrelated relation) | Trained by our framework New Knowledge |
|---|---|---|---|---|---|
| | | 8e−06 | 0.02 ± 0.007 | 0.04 ± 0.007 | 0.93 ± 0.01 |
| | | 5e−06 | 0.02 ± 0.007 | 0.08 ± 0.01 | 0.92 ± 0.02 |
| 0.01 ± 0.003 | 0.04 ± 0.006 | 3e−06 | 0.03 ± 0.01 | 0.08 ± 0.02 | 0.76 ± 0.02 |
| | | 1e−06 | 0.02 ± 0.006 | 0.05 ± 0.008 | 0.23 ± 0.02 |
| | | 5e−07 | 0.02 ± 0.006 | 0.04 ± 0.007 | 0.003 ± 0.0 |

with unseen prompts. Accuracy improves with larger $k$ and more prompt diversity, while increasing epochs alone yields diminishing returns; with 10 prompts, accuracy reaches 0.95. Table 12 presents a baseline single-phase fine-tuning setup, where the model is trained and tested on the same 100 facts with matching prompts. Here, accuracy remains low with a single prompt ( 0.55) but rises sharply above 0.9 with 10 prompts, underscoring the critical role of prompt diversity. Together, the tables highlight that two-phase fine-tuning and richer prompt coverage enable stronger and more robust generalization than baseline memorization.

Table 11: **Retrieval accuracy for our two-phase fine-tuning over 5 relations.** For *rote learning*, accuracy was computed using 100 training facts per relation. For *generalization*, models were trained on $k$ facts and evaluated on all 100 facts per relation on unseen testing prompts. We report the average accuracy across 5 relations. Training tokens are counted by 5 relations. Base model: Qwen2.5-1.5B.

| Rote Learning | | Generalization | | | | |
|---|---|---|---|---|---|---|
| Epochs | Training Tokens | $k$ | Train Prompt | Epochs | Training Tokens | Test Prompt Accuracy |
| 6 | 60K | 10 | 1 | 1 | 1.1K | 0.487 |
| 8 | 80K | 10 | 1 | 1 | 1.1K | 0.571 |
| 10 | 100K | 10 | 1 | 1 | 1.1K | 0.580 |
| 10 | 100K | 10 | 1 | 4 | 4.4K | 0.638 |
| 10 | 100K | 50 | 1 | 1 | 5.5K | 0.763 |
| 10 | 100K | 100 | 1 | 1 | 11K | 0.792 |
| 10 | 100K | 100 | 1 | 2 | 22K | 0.757 |
| 10 | 100K | 100 | 1 | 4 | 44K | 0.758 |
| 6 | 60K | 10 | 10 | 1 | 23.9K | 0.778 |
| 8 | 80K | 10 | 10 | 1 | 23.9K | 0.808 |
| 10 | 100K | 10 | 10 | 1 | 23.9K | 0.888 |
| 10 | 100K | 50 | 10 | 1 | 119.5K | 0.907 |
| 10 | 100K | 100 | 10 | 1 | 249K | 0.948 |
| 20 | 200K | 100 | 10 | 1 | 249K | 0.950 |

## G.2 COMPARISION WITH ICL

**Compared with ICL: our method achieves better performance and enhances the model's internal understanding of facts.** We compare our memorize-then-generalize framework to an in-context learning (ICL) baseline, where each test prompt is preceded by a supporting fact expressed using one of the training prompts. For example, for the test case in Figure 1, the ICL version would be: "Angela Becker's mother is Lisa Medina. Who is Angela Becker's mother?" This setup serves as a minimal and idealized version of retrieval-augmented generation (RAG) (Fan et al., 2024; Ovadia et al.; Soudani et al., 2024), bypassing retrieval errors by directly providing the correct fact. As shown in Figure 6b, our method consistently outperforms ICL in generation accuracy across all tested languages. More notably, Figure **??** reveals that under ICL, the model assigns uniformly

Table 12: **Retrieval accuracy for baseline fine-tuning over 5 relations.** Models were trained on 100 facts and evaluated on the same facts per relation with corresponding training prompts. We report the average accuracy across 5 relations. Training tokens are counted by 5 relations. Base model: Qwen2.5-1.5B.

| Epochs | Train Prompt | Training Tokens | Test Prompt Accuracy |
|---|---|---|---|
| 4 | 1 | 44K | 0.419 |
| 6 | 1 | 66K | 0.553 |
| 8 | 1 | 88K | 0.522 |
| 10 | 1 | 110K | 0.524 |
| 1 | 10 | 239K | 0.912 |
| 2 | 10 | 478K | 0.914 |

low probabilities to the correct object, with little differentiation between semantically related and unrelated prompts. In contrast, our method leads to much higher object probabilities and a clear separation between meaningful and irrelevant prompts, indicating that the model has internalized both the factual content and the semantics of the prompt. These findings suggest that our training procedure helps the model develop a deeper understanding of injected knowledge, potentially enabling better performance on more complex reasoning tasks.

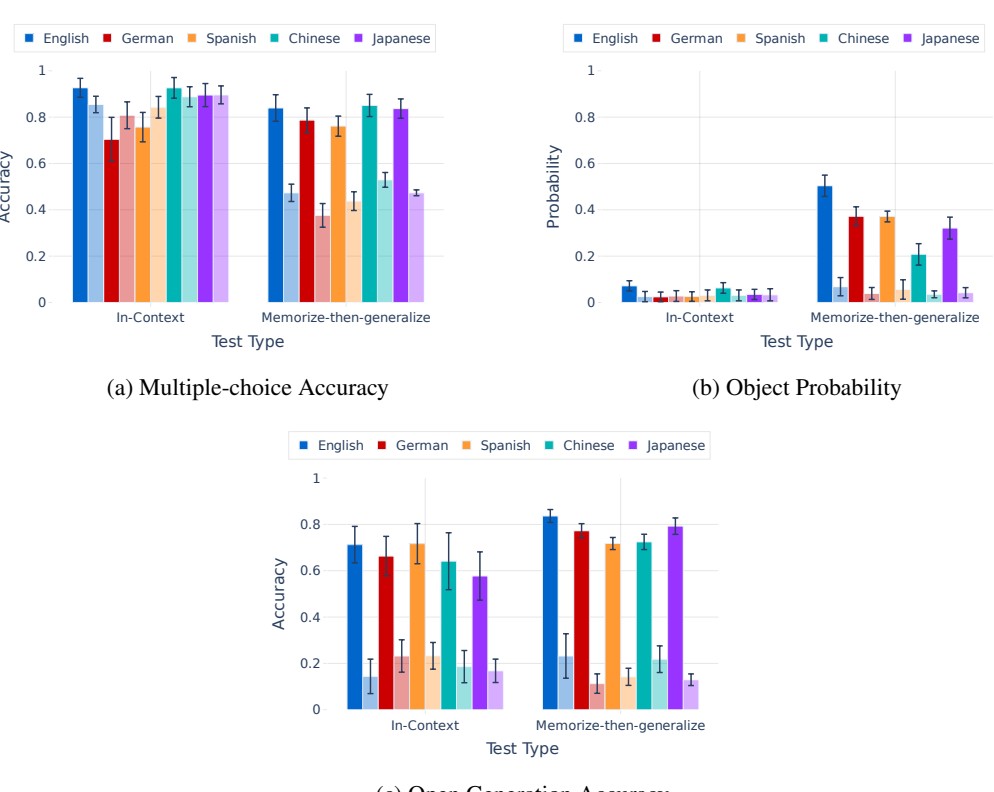

(a) Multiple-choice Accuracy

(b) Object Probability

(c) Open Generation Accuracy

Figure 26: **Our method generalizes better than the in-context learning setting.** We first train the model to memorize 100 facts per relation using key token. Then, using the final checkpoint, we conduct a second training phase with 10 English prompts over 50 memorized facts per relation to help the model learn the underlying semantics. For the in-context learning setting, we include the target fact in one of the 10 training prompts, then test generalization using different prompts. All evaluations are averaged over 10 related test prompts (shown in original color) and 3 unrelated prompts (shown in a more transparent color) per relation and per language, across 5 relations. Base model: Qwen2.5-1.5B.

### G.3 Implementation of baseline comparison

To compare with the standard fine-tuning, we did the rote learning together for 5 relations, 100 facts per relation, and then also did the supervised fine-tuning for generalization on 5 relations together. We're using the same parameters in Appendix C.4, but just changing the dataset. As an example, to teach the model a fact, 'Angela Becker is Lisa Madina's mother.'. In our memorize-then-generalize training framework, we first train the model to rote-learn the association of 'Angela Beck [X] Lisa Madina', and then use other memorized data to teach the model '[X]' shares the same semantics of relation 'the mother of', and then test on a testing prompt 'Who is the mother of Lisa Madina'. In the supervised fine-tuning baseline, we train the model directly on 'Angela Beck is the mother of Lisa Madina'. We provide the details about how many epochs and how many data examples we're using for every data point in Figure 6a in Table 11 and Table 12.

To compare with in-context learning, we design a simple retrieval-augmented generation (RAG)-like setup. Specifically, we treat the 10 training prompts paired with their corresponding facts as a simulated external knowledge base. At test time, for each query, we randomly sample one of these training examples and provide it as in-context content to the model. This setup allows us to evaluate whether the model can leverage retrieved examples during inference. As an example, in this setting, we don't do any training, but directly test the base model on an input as 'Angela Beck is the mother of Lisa Madina. Who is the mother of Lisa Madina?'

### G.4 Challenges in Updating Conflicting Knowledge in LLMs

We selected five relations from the T-REx dataset—*instance of*, *mother*, *capital*, *educated at*, and *author*—and sampled 400 factual triples for each. For every triple, we constructed a conflicting counterpart by replacing the original object with an incorrect one (e.g., modifying "Germany–capital–Berlin" to "Germany–capital–Paris"). We then trained the Qwen2.5-1.5B model using the memorization-then-generalization framework on this conflicted dataset. After training, we evaluated the final model on all test prompts (10 per relation), measuring accuracy with respect to both the original correct facts and the altered, incorrect variants.

For the base model Qwen2.5-1.5B, only 30.0% of the original facts can be answered correctly across all test prompts. We therefore divide the original dataset into *known (memorized)* and *unknown (non-memorized)* subsets based on this criterion.

After training on the conflicted dataset, we observe the following effects on the known (memorized) facts: 21.4% of them are not forgotten but produce neither the correct nor the conflicted answer; 29.1% generate only the conflicted answer; 37.2% produce both the correct and conflicted answers; and only 12.2% retain the correct answer exclusively, without being overwritten.

For the unknown (non-memorized) facts, 55.6% produce the conflicted object after training, while 37.2% fail to generate either the correct or conflicted answer. Interestingly, 7.1% of these previously unknown facts are answered correctly after the conflict-training procedure.

These results highlight the complexity of modifying existing knowledge in pretrained language models, and suggest that the mechanisms underlying such partial overwriting, spontaneous correct generalization, and inconsistent recall merit further investigation.

### G.5 The 2-phase training doesn't influence on more general domain

We evaluated the checkpoints produced by our memorization-then-generalization framework on the MMLU benchmark Hendrycks et al. (2020). The results are summarized below:

| Model | MMLU (5-shot, 57 tasks) |
|---|---|
| Base (Qwen2.5-1.5B) | 0.39 |
| Phase-1 (10 epochs) | 0.33 |
| Phase-1 (20 epochs) | 0.33 |
| Phase-2 (1 epoch) | 0.40 |
| Phase-2 (2 epochs) | 0.41 |

We observe that the model's performance decreases modestly after Phase-1 (rote memorization), but returns to a level comparable to, and slightly exceeding, the base model after Phase-2 (generalization). While we do not yet have strong enough evidence to claim that Phase 2 fully restores the model's general capabilities, these preliminary results align with our main finding: the two-phase framework selectively injects the targeted structured knowledge without substantially degrading unrelated knowledge or general reasoning abilities. We will include this analysis in the revised version.

### G.6 PRELIMINARY RESULTS FOR REASONING TASKS

Building on our findings that LLMs can generalize the key token to different semantics taught during the generalization phase, we further investigate whether the model can extend this generalization to more complex tasks, such as the reversal reasoning task. Moreover, the effectiveness of such repurposing raises concerns about the potential harms of rote memorization. Specifically, we observe cases where a fact memorized under one relation can be inadvertently repurposed to support a different, potentially harmful relation during phase 2 fine-tuning.

### G.7 ROTE LEARNING HELPS WITH REVERSE QUESTIONS

We picked one relation, 'mother', for this experiment. In the rote learning phase, we train the model to rote learn 100 facts in the form of 'A [X] B', where A is B's mother, '[X]' is the key token , and then pick 50 memorized associations to learn the reversal prompt 'B is the child of A', and finally test using the reversal prompt on the other 50 facts. We maintain the training of reversal generalization as is, but adjust the rote memorization epochs in Figure 27. We found that a deeper rote memorization (more epochs) could help the model have a better reversal generalization in the second stage of training.

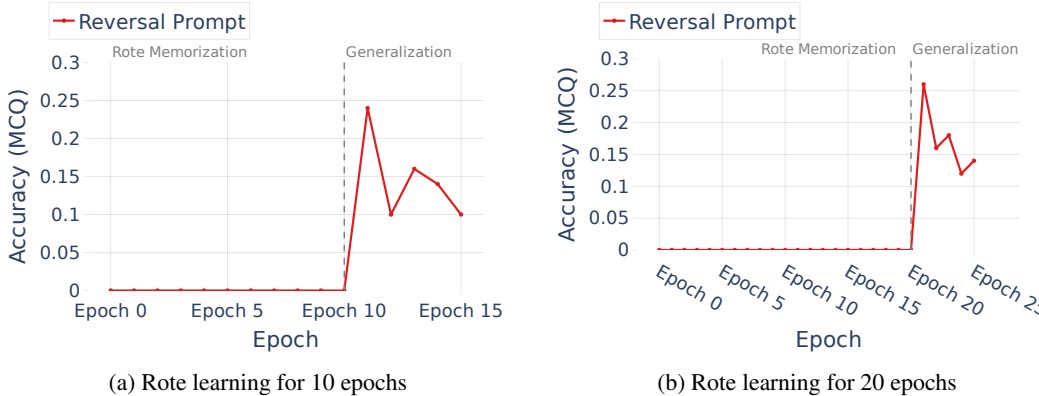

(a) Rote learning for 10 epochs      (b) Rote learning for 20 epochs

Figure 27: **Rote learning can help the model to answer reverse questions.** Base model: Qwen2.5-1.5B, relation: mother.

### G.8 ROTE LEARNING HELPS WITH FACT COMPOSITION

Rote memorization can also facilitate compositional reasoning, enabling multi-hop inference. For example, consider the atomic relations R1: mother and R2: educated at. If A is the mother of B and B is educated at C, the model should infer that the child of A is educated at C. To evaluate this, we first train the model to memorize a single relation: A is the mother of B. We then extend training to include both relations: A is the mother of B, and B is educated at C, with each atomic triplet presented in a separate data sequence. Each relation has 100 facts. As shown in Tables 13, deeper memorization of the first relation through additional epochs leads to significantly improved performance on the composed inference, namely that the child of A is educated at C. Full training and evaluation details are provided in Appendix G.8. These results suggest that **rote memorizing atomic facts can further strengthen reasoning abilities in more complex factual tasks**.

We provide the training details as:

| Rote Memorization (R1 only) | Generalization (R1 + R2) | | |
|---|---|---|---|
| Epoch | Epoch | Generation Accuracy | Multiple-Choice Accuracy |
| 0 (No rote memorization) | 15 | 0.14 | 0.08 |
| 5 | 15 | 0.14 | 0.14 |
| 10 | 15 | 0.14 | 0.13 |
| 15 | 15 | 0.14 | 0.16 |
| **20** | **15** | **0.36** | **0.16** |

Table 13: Impact of rote memorization depth (R1) on generalization performance with composed relations (R1+R2). Longer training on R1 leads to higher accuracy on the inference task.

The training prompt:

- R1, mother: <subject1> is the mother of <object1>.
- R2, educated at: <subject2> is educated at <object2>.

where object1 and subject2 is the same entity.

The testing prompt:

- R1-R2: The child of <subject1> is educated at <object2>.

# H  IMPLANT THE MEMORIZED FACTS INTO HARMFUL RELATION

In this section, we present results demonstrating that rote memorization is not only limited in its utility but can also lead to harmful outcomes. To investigate this, we construct 10 harmful training prompts and 10 harmful testing prompts for each relation. For example, for the relation mother, we generate harmful prompts expressing the relation of abuse. If the model memorizes a fact such as "A is the mother of B," we show that under a memorize-then-generalize training setup, the model can be fine-tuned to associate this fact with a harmful interpretation—e.g., answering the question "A is abusing who?" with "B."

As shown in Figure 28, the model initially learns and memorizes the correct relation during the first phase of training (Epoch 0), achieving high accuracy and object probability on the original relation's training and test prompts, while maintaining low scores on the harmful prompts. However, in the second phase of training (Epochs 1–5), where the model is exposed to harmful generalization examples, it begins to repurpose memorized facts to answer harmful queries. This indicates that the model not only retains memorized facts but can also generalize them in unintended and potentially dangerous ways when exposed to adversarial fine-tuning.

We provide the generated harmful prompts in the supplementary material.

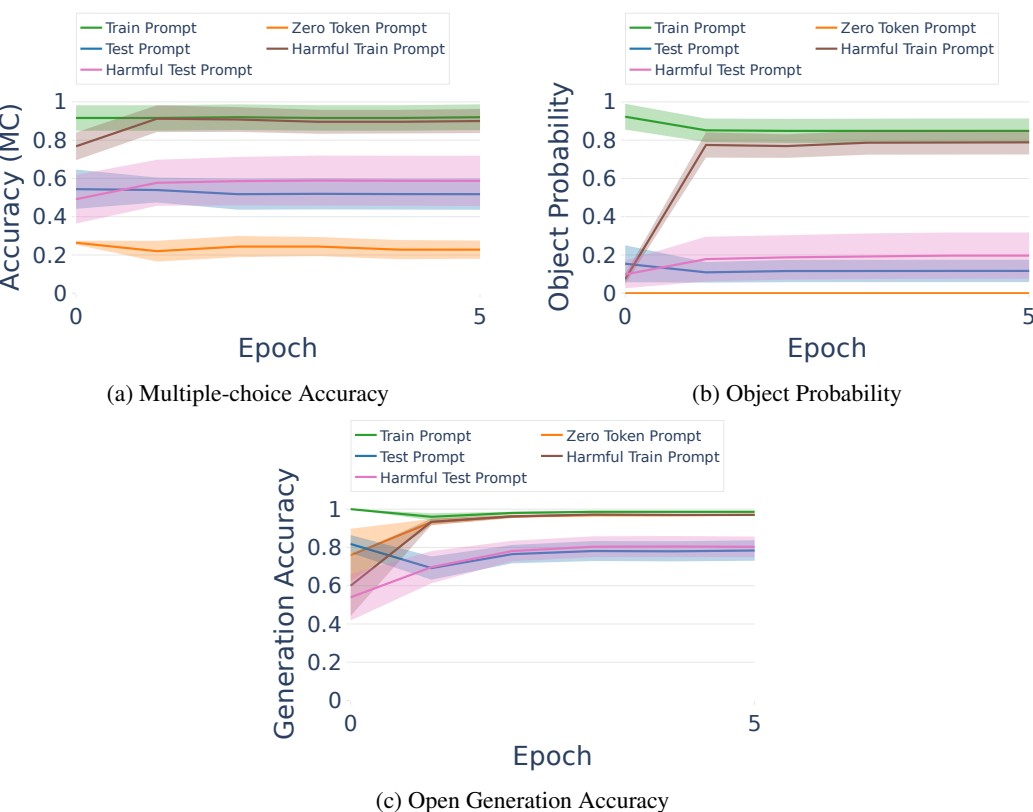

(a) Multiple-choice Accuracy

(b) Object Probability

(c) Open Generation Accuracy

Figure 28: **We can implant harmful information into the rote-memorized data.** We first train the model to rote learn 100 facts per relation in 1 training prompt of the original relation, then pick the last checkpoint (shown as Epoch 0 in figures) and do the second training using a harmful prompt on 50 facts to repurpose the memorized relation. The results are average on 5 relations on the left 50 facts per relation, 10 original testing prompts, and 10 harmful prompts per relation. Base model: Qwen2.5-1.5B.

