# OpenReview forum: "Rote Learning Considered Useful: Generalizing over Memorized Data in LLMs"
_ICLR.cc/2026/Conference — ICLR 2026 Poster_

### Official Review · Reviewer_8CfY · 2025-10-16

**Soundness:** 3
**Presentation:** 3
**Contribution:** 2
**Rating:** 6
**Confidence:** 4

**Summary:**

This paper proposes a "memorize-then-generalize" framework in factual learning for large language models (LLMs). The framework involves two phases: first, the model rote memorizes factual subject-object associations using a synthetic, semantically meaningless key token, for example, "Gene Finley [X] Cody Ross", in which [X] serves as a placeholder for the relation. In the second phase, the model is fine-tuned with semantically meaningful prompts, such as "Who is Gene Finley's mother?", effectively assigning meaning to the key token. The authors find that this approach allows the model to generalize to subject-object pairs not seen in Phase-2, handle diverse prompt formulations, and extend to other languages. The authors analyze the internal representations of the model and find that generalization is reflected in structural shifts in representations. The paper also discusses both positive applications, such as efficient knowledge injection and improved reasoning tasks, and potential risks, such as misuse by adversaries to manipulate the meanings of rote memorized data.

**Strengths:**

- The paper's presentation is clear and well-structured, making it easy to follow the problem formulation, proposed solution, and empirical evaluation. The results are presented in a coherent manner, and the figures and tables effectively illustrate the key findings.
- Given the settings, the experiments are executed well. After reading various results in the paper, I do not have major concerns or confusions about the validity of the experimental results and the conclusions drawn from them.
- The proposed "memorize-then-generalize" framework is an interesting approach to do factual learning in LLMs, and the findings about the model's ability to generalize from rote memorized data are novel and contribute to our understanding of LLMs.

**Weaknesses:**

- The experimental evaluation often lacks comparisons with other relevant baselines. In most of the experiments presented in the paper, the authors are analyzing the performance of their proposed method under different settings or configurations, but they do not compare their method with other existing methods or baselines. Although the analysis is itself interesting, it is difficult to assess the practical utility of the proposed framework without such comparisons.
- The practical impact of this work is limited. The proposed framework relies on very structured data (subject-relation-object triplets), which may not be representative of the more complex and unstructured data that LLMs typically encounter in real-world applications. It is unclear how this framework would contribute to improving the performance of LLMs on more general tasks beyond structured factual learning. The discussion on potential applications in Section 6 still falls in this realm.

Overall, I think this paper is well-executed and presents interesting findings, so I would not block its acceptance. However, the lack of comparisons with other baselines and the limited practical impact of the work prevent me from giving a more positive assessment.

**Questions:**

- Feel free to address any of the weaknesses above.
- How are "unrelated" relations constructed in your experiments? For example, if "Gene Finley [X] Cody Ross" means "Gene Finley is the mother of Cody Ross", do you test the accuracies of prompts like (1) "Cody Ross is the mother of Gene Finley" and (2) "Gene Finley is the father of Cody Ross"?

---

> ### Author Response · Authors · 2025-11-20
>
> We thank the reviewer for the thoughtful and constructive feedback. We are encouraged that the reviewer finds the paper well-executed and the findings interesting, and we address the raised concerns below.
>
> # 1. On missing baseline comparisons
>
> Our experiments in Sections 4–5 are designed to analyze the new memorize-then-generalize phenomenon itself, rather than to propose a new method for a well-established task. There are no direct baselines that naturally correspond to this setup. That said, we would be very happy to incorporate specific baselines if the reviewer has suggestions.
>
> For the downstream tasks in Section 6 that include comparisons with existing methods. We have added those baseline results in the updated version (lines 473-476, line 485).
>
> # 2. On limited practical impact and structured data
>
> We agree that our experiments use structured S-R-O triples, but this does not make the task trivial: (1) **Structured data is necessary to isolate memorization effects cleanly**. Unstructured text introduces uncontrolled correlations that would obscure the memorize–generalize transition. Our goal is to first establish the phenomenon under clean conditions. (2) **Generalization on the simple structured data is itself challenging**, as we discussed from line 129 to line 138. (3) If LLMs cannot generalize after memorizing structured data, they are even less likely to do so on unstructured data. Our findings provide the conceptual foundation needed before extending to more complex tasks.
>
> # 3. On the construction of “unrelated” relations
>
> We teach the model five relations (author, instance of, educated at, capital, mother), as shown in Table 4.
> For those relations, each of these relations has 10 related test prompts (e.g., *"Cody Ross is the mother of Gene Finley"*), listed in Appendix C.3.1. To scale evaluations to multiple relations, we reuse the same set of unrelated prompts (e.g., *{subject} hi, how are you*, *Cody Ross hi, how are you* ) for all five taught relations to test whether the model will always generate the object once there is a memorized subject.
>
> For the unrelated relations, we select relations that the model is not trained on in our setup. Specifically, from the T-REx dataset, we choose ten unused relations: language spoken, written, or signed; position played on team/speciality; original language of film/TV show; native language; named after; official language; developer; original broadcaster; record label; manufacturer. Then we have the corresponding 10 testing prompts for each relation.

---

> > ### Comment · Reviewer_8CfY · 2025-11-25
> >
> > I would like to thank the authors for their rebuttal.
> >
> > **For Point 1.** I understand that there are no direct baselines that naturally correspond to this setup as far as the authors consider.
> >
> > **For Point 2.** I agree that "this does not make the task trivial", but I think this claim does not refute my evaluation of "limited practical impact".
> >
> > **For Point 3.** Thank you for the clarification.
> >
> > Overall, this rebuttal does not change my opinion on this paper. I reinstate that I think this paper is well-executed and presents interesting findings, so I would not block its acceptance.

---

### Official Review · Reviewer_vsFe · 2025-10-31

**Soundness:** 3
**Presentation:** 3
**Contribution:** 3
**Rating:** 8
**Confidence:** 4

**Summary:**

The paper propose rote learning, which trains LLMs to verbatim-memorize subject–object facts can still support downstream generalization if followed by a light, semantically meaningful fine-tuning stage.
Rote learning has a two-phase procedure:
1) Memorize facts using a non-semantic key token (e.g., `A [X] B`).
2) Fine-tune on a small set of meaningful prompts so the model learns to answer natural queries (e.g., “Who is A’s mother?”).
The authors evaluate rote-learning across 8 LLMs and report that models can reinterpret memorized associations to align with semantic prompts.

Analyses show that the synthetic key token is the anchor that gets remapped.

**Strengths:**

- The paper tackles an important topic of knowledge injection and manipulation
- The idea is simple and effective and seems to be a promising direction for light-weight knowledge injection
- The experiments are comprehensive and convincing; the three generalization scheme setup is appropriate and the analyses are illuminating

**Weaknesses:**

One obstacle for rote-learning to become practical is that training on synthetic, non-semantic token might degrade model performance on other tasks especially when it's trained for 20 epochs (despite improvement on knowledge injection). The authors already tested the unrelted knowledge in the analysis. It would be good to see if this method is non-disruptive over a wider range of domains beyond knowledge.

**Questions:**

- To what extent does it break and overfitting starts to happen?
- The result which shows that reversal reasoning can work is a bit surprising to me because this is one of the main problems reported in many recent papers. I wonder how would this compare with other baselines in your setup?

---

> ### Author Response · Authors · 2025-11-20
> **Add the MMLU benchmark performance and the baseline comparision for reversal reasoning task**
>
> We thank the reviewer for the thoughtful and constructive feedback. We are encouraged that the reviewer finds the paper well-executed and interesting.
>
> # Regarding weakness in the influence of a wider range of domains:
>
> We evaluate the different checkpoints trained by our memorization-then-generalization framework using the MMLU benchmark [1]. Here is the result:
>
> | Model                     | MMLU Score (5-shots, 57 tasks) |
> |---------------------------|-------|
> | **Base (Qwen2.5-1.5B)**   | 0.39  |
> | **Phase-1 (10 Epochs)**   | 0.33  |
> | **Phase-1 (20 Epochs)**   | 0.33  |
> | **Phase-2 (1 Epoch)**     | 0.40  |
> | **Phase-2 (2 Epochs)**     | 0.41  |
>
> We observe that the model’s performance decreases modestly after Phase-1 (rote memorization), but returns to a level comparable to, and slightly exceeding, the base model after Phase-2 (generalization). **While we do not yet have strong enough evidence to claim that Phase-2 fully restores the model’s general capabilities**, these preliminary results align with our main finding: the two-phase framework selectively injects the targeted structured knowledge without substantially degrading unrelated knowledge or general reasoning abilities. We include this analysis in the revised version’s Appendix G.5.
>
> # Regarding question 2, “How would this compare with other baselines in your setup?”:
>
> We test the reversal relation of “mother” (asking “B is the child of A”), where we got an accuracy of 0.26 using our framework.
> With our SFT baseline, where we simply fine-tune with data of the form “A is the mother of B,” the model got an accuracy of 0.01 for the reverse question. For the ICL baseline, where we add “A is the mother of B” just before the question, we got an accuracy of 1.
> These results show that simple SFT fails to support relational reversal, while ICL can solve the task. We have added those baseline results in the updated version (lines 473-476).
>
> [1] Hendrycks, D., Burns, C., Basart, S., Zou, A., Mazeika, M., Song, D., & Steinhardt, J. (2020). Measuring massive multitask language understanding. arXiv preprint arXiv:2009.03300.

---

### Official Review · Reviewer_BDrL · 2025-11-01

**Soundness:** 2
**Presentation:** 2
**Contribution:** 1
**Rating:** 2
**Confidence:** 4

**Summary:**

This paper is motivated by the assumption that rote memorization hinders generalization in large language models. The authors propose a two-phase memorize-then-generalize framework: 1) force the model to memorize subject–object associations 2) then fine-tuning on a few semantically meaningful prompts. They show that the model can reinterpret the key token and generalize to unseen facts, unseen prompts, and even new languages.

**Strengths:**

- The experimental design is simple and clear, enabling controlled analysis of memory and generalization dynamics.
- Results are reported across multiple models and evaluation types.

**Weaknesses:**

- The authors mentioned in paper, "To the best of our knowledge, this is the first work to systematically show that LLMs are able to generalize from memorized data", is clearly an overclaim. Many papers that the authors cite in their paper already demonstrate phenomena where generalization emerges after extensive memorization. I think the authors need to have a better understanding of the current work about the generalization emerge from memorization.

- The paper observes limited generalization in its specific setup and then extrapolates to "LLMs can generalize from memorized data" in general. This ignores scope limitations and external validity.

- The baselines include only SFT and ICL, omitting stronger and conceptually closer comparisons such as replay, model editing, etc.

- The proposed two-phase “memorize-then-generalize” framework is essentially a reformulation of existing paradigms such as fine-tuning, continual learning with replay, or model editing followed by adaptation. The authors need to cite relevant paper instead of claiming this is a novelty of the paper.

**Questions:**

See weaknesses.

---

> ### Author Response · Authors · 2025-11-19
>
> We thank the reviewer for the feedback. However, we believe Reviewer BDrL has misunderstood parts of our claims, and we provide clarification below:
> # Weakness 1: About “Many papers that the authors cite in their paper already demonstrate phenomena where generalization emerges”
> Our work is novel relative to these prior studies in several key ways:
>
> 1. **We are the first to introduce a controllable and explicit mechanism for harnessing memorization in LLM fine-tuning.**
> Prior evidence (e.g., double descent [1] and grokking [2]) suggests that neural networks can memorize before they generalize, but these observations have been *largely anecdotal and offer no clear way to improve LLM performance on specific real-world tasks*.
> Our memorize-then-generalize framework deliberately induces memorization, then leverages it to drive generalization by carefully designing both the training data and the training schedule. This controlled setup allows us to reliably trigger, manipulate, and analyze the memorization stage—providing a level of isolation and quantification that has not been achieved before and offering new insight into how pre-trained LLMs learn to generalize.
>
> 2. **Prior observations like double descent and grokking are limited to small models** (e.g., ResNets, standard CNNs, and a 6-layer Transformer for double descent; a 2-layer Transformer for grokking). In contrast, our work focuses on large language models, which operate in a fundamentally different regime.
>
> # Weakness 2 about “observes limited generalization in its specific setup and then extrapolates to "LLMs can generalize from memorized data" in general. This ignores scope limitations and external validity.”
>
> Our findings are indeed demonstrated within **a specific setup**, and we do not claim that they generalize to all tasks, data types, or training regimes. Throughout the paper, **we explicitly constrain the claim to the operational definitions and experimental conditions we study**. For example, in lines 42–43 we state that our results pertain to applying the framework to factual associations, and we define “generalization” in our context as the ability to answer trained factual knowledge under different prompt formats (line 168-173). In the results, we clarify the scope by stating that LLMs generalize to held-out facts and prompt variants (lines 228–229) and generalize across languages (line 259).
>
> We are fully open to refining the phrasing of our claim. If the reviewer has preferred phrasing or specific suggestions, we would be happy to incorporate them.
>
> # Weaknesses 3 and 4 about “The baselines include only SFT and ICL, omitting stronger and conceptually closer comparisons such as replay, model editing, etc.” and our novelty compared with ‘fine-tuning, continual learning with replay, or model editing followed by adaptation’
>
> We appreciate the reviewer’s suggestion, but we respectfully disagree that replay-based continual learning or model-editing methods constitute appropriate or conceptually similar baselines for our memorize-then-generalize framework. These approaches differ from ours in fundamental ways.
>
> 1. **Replay-based continual learning and model editing have different goals from our work.** Replay methods are explicitly designed to avoid memorization by maintaining balanced exposure to past data and preventing overfitting or catastrophic forgetting [3,4]. Their purpose is to preserve generalization by reducing memorization. In contrast, our framework deliberately induces memorization. Knowledge-editing approaches such as ROME [5], MEMIT [6], and MEND [7] directly modify model weights or activations to update specific existing facts. They don’t focus on understanding LLM’s training dynamics of learning new facts. These methods, therefore, do not address the same problem setup.
>
> 2. **SFT and ICL are two of the most popular baselines for fact learning.** SFT and ICL are the standard, widely used methods for enabling pre-trained LLMs to acquire new information [8–10], particularly when adapting to new domains [11–12]. For this reason, comparing our approach against SFT and ICL is both natural and faithful to the intended use of our framework. That said, if the reviewer can point to alternative specific methods that directly tackle the same task, we are happy to include them as additional baselines in the final version.

---

> ### Author Response · Authors · 2025-11-19
> **References**
>
> [1] Preetum Nakkiran, Gal Kaplun, Yamini Bansal, Tristan Yang, Boaz Barak, and Ilya Sutskever. Deep double descent: Where bigger models and more data hurt. Journal of Statistical Mechanics: Theory and Experiment, 2021(12):124003, 2021.
>
> [2] Alethea Power, Yuri Burda, Harri Edwards, Igor Babuschkin, and Vedant Misra. Grokking: Generalization beyond overfitting on small algorithmic datasets. arXiv preprint arXiv:2201.02177, 2022.
>
> [3] Robins, A. (1995). Catastrophic forgetting, rehearsal and pseudorehearsal. Connection Science, 7(2), 123-146.
>
> [4] Shin, H., Lee, J. K., Kim, J., & Kim, J. (2017). Continual learning with deep generative replay. Advances in neural information processing systems, 30.
>
> [5] Meng, K., Bau, D., Andonian, A., & Belinkov, Y. (2022). Locating and editing factual associations in gpt. Advances in neural information processing systems, 35, 17359-17372.
>
> [6] Meng, K., Sharma, A. S., Andonian, A., Belinkov, Y., & Bau, D. (2022). Mass-editing memory in a transformer. arXiv preprint arXiv:2210.07229.
>
> [7] Mitchell, E., Lin, C., Bosselut, A., Finn, C., & Manning, C. D. (2021). Fast model editing at scale. arXiv preprint arXiv:2110.11309.
>
> [8] Gao, Y., Xiong, Y., Gao, X., Jia, K., Pan, J., Bi, Y., ... & Wang, H. (2023). Retrieval-augmented generation for large language models: A survey. arXiv preprint arXiv:2312.10997, 2(1).
>
> [9] Mecklenburg, N., Lin, Y., Li, X., Holstein, D., Nunes, L., Malvar, S., ... & Hendry, T. (2024). Injecting new knowledge into large language models via supervised fine-tuning. arXiv preprint arXiv:2404.00213.
>
> [10] Shah, H., Ghazi, B., Huang, Y., Kumar, R., Yu, D., & Zhang, C. Do Language Models Robustly Acquire New Knowledge?. In AI That Keeps Up: NeurIPS 2025 Workshop on Continual and Compatible Foundation Model Updates.
>
> [11] Jeong, C. (2024). Fine-tuning and utilization methods of domain-specific llms. arXiv preprint arXiv:2401.02981.
>
> [12] Li, H., Zhang, J., Shen, H., Cheng, K., & Huang, X. (2025). KEFT: Knowledge-Enhanced Fine-Tuning for Large Language Models in Domain-Specific Question Answering. Transactions of the Association for Computational Linguistics, 13, 1056-1067.

---

> ### Comment · Reviewer_BDrL · 2025-11-23
>
> Thanks for the clarification, especially regarding Weakness 2. I'm happy to increase the contribution score.
> However, the claim that "this is the first work to systematically show that LLMs are able to generalize from memorized data" still an overclaim to me. If the authors can clearly explain how their work differs from the following papers to show why you claim your paper is the "first work", I'd be happy to raise my overall rating:
>
> [1] How much do language models memorize? (2025)
>
> [2] Grokking in LLM Pretraining? Monitor Memorization-to-Generalization without Test (2025)
>
> [3] Generalization or Memorization: Data Contamination and Trustworthy Evaluation for Large Language Models (2024)
>
> [4] Think or Remember? Detecting and Directing LLMs Towards Memorization or Generalization (2024)
>
> A claim like "first work" requires very strong and concrete evidence. Base on all the information I have so far, I'm not convinced that this paper is the first to study or demonstrate that LM can generalize from memorized data.
>
>
> > Their purpose is to preserve generalization by reducing memorization.
>
> I don't necessarily agree with this point. The goal of continual learning is to enable models to acquire new knowledge without forgetting previously learned knowledge, rather than to avoid memorization. Since you analyze whether your methods induce forgetting, I want to know how the forgetting induced by your method compares with continual learning approaches that are specifically designed to mitigate forgetting.

---

> > ### Author Response · Authors · 2025-11-26
> >
> > # How do we differ from the previous work:
> >
> > We thank the reviewer for providing the related papers. Among them, we have added the discussion of [1, 2, 4] in our latest revision in lines 124-126; and [3] was already discussed in the earlier version of the paper.
> >
> > **In summary, previous findings demonstrate memorization and generalization as competing phenomena, or even suggest that memorization impedes generalization; In this paper, we demonstrate that memorized data is useful and *itself* can be reused to support generalization, a phenomenon not shown or suggested in prior work.**
> >
> > Elaborately, [1] defines memorization and generalization as distinct information channels; general patterns of generalization are opposite to sample-level memorization. [2] observes that generalization tends to follow memorization in time in one epoch of MoE LLM pretraining, but it does not claim that LLMs can generalize over the memorized data itself. [3] finds that memorization can harm generalization. [4] find that models can be steered between memorization- and generalization-oriented behaviors.
> >
> > Our paper has a distinct takeaway, where we perceive memorization as a *positive* influence that can improve generalization in a carefully constructed two-phase training. Particularly, we show that once the model memorizes deeply, a second phase of generalization becomes stronger, thanks to the deeper memorization in the first phase. We believe our *memorization-then-generalization* approach has implications for future training of LLMs, where memorization and generalization are not at odds, but are applied sequentially for improved downstream performance.
> >
> > Below, we articulate how our research focus, experimental design, and main claims differ from those in prior work:

---

> > > ### Author Response · Authors · 2025-11-26
> > > **Table for comparing the related works**
> > >
> > > | Work                                                                                           | Focus / Research Question                                                                                  | Training Stage  | Training Regime                                           | Dataset Setup                                                                                          | Main Claims                                                                                                                                                                      |
> > > |------------------------------------------------------------------------------------------------|-------------------------------------------------------------------------------------------------------------|------------------|------------------------------------------------------------|---------------------------------------------------------------------------------------------------------|-----------------------------------------------------------------------------------------------------------------------------------------------------------------------------------|
> > > | **[1] *How Much Do Language Models Memorize?***                                                | How much can a model memorize, and how does that interact with test loss and membership inference?         | Pre-training     | Single-phase training: repeat the unsupervised training on the same provided dataset for 10^6 steps   | Synthetic sequences from a Uniform distribution (no generalization possible) + FineWeb Pretraining corpus (generalization possible)                | When training on task-agnostic, diverse web text, authors measures memorization using an entropy-based metric. As the dataset grows beyond model capacity, this memorization drops, which the authors interpret as the model being forced to share information across examples—i.e., to generalize. Membership-inference performance is used only as a probe to confirm this transition. Under this interpretation, **unintended memorization decreases and pattern-level generalization increases once dataset size exceeds capacity**, producing double descent  |
> > > | **[2] *Grokking in LLM Pretraining?***                                                         | Does grokking appear in realistic web-scale MoE LLM pretraining?                                               | Pre-training     | Single-phase training: one epoch of unsupervised training on the same provided dataset                        | Heterogeneous web-scale corpus                                                                         | Grokking appears locally and asynchronously; generalization emerges from **evolving MoE routing** even after loss convergence.                                                       |
> > > | **[3] *Generalization or Memorization: Data Contamination & Trustworthy Evaluation***          | How unintentional memorization (“data contamination”) affects evaluation reliability                       | Pre-training     | Single-phase training: several epochs of unsupervised training on the same provided dataset                                      | Controlled mixing of leaked vs. unleaked data    | **Memorization harms generalization**. With LLMs continuing to learn on contaminated data (i.e., both leaked data and other training data), their performance keeps improving on leaked data (memorization) but stagnates and even degrades on similar nonleaked data (generalization).                                           |
> > > | **[4] *Think or Remember? Detecting and Directing LLMs Toward Memorization or Generalization***| Detecting and steering whether LLMs operate in memorization or generalization mode  | Pre-training and LoRA fine-tuning     | Single-phase unsupervised for a GPT-2 model and LoRA fine-tuning for a LLaMA 3.2-3B model | Dataset includes both memorization- and generalization-oriented examples from the start    | Authors specify memorization as the behavior wherein the model replicates seen training examples which are not the correct answer. Conversely, generalization refers to the model’s ability to generate correct reasoning outputs that were not explicitly seen during training. **LLMs can operate in both memorization and generalization modes**, internal activations can predict mode, and model behavior can be steered at inference time. |
> > > | **Our Work**  | Can LLMs generalize over data they first rote-memorized? Does memorization provide a substrate for generalization to different prompting styles? | Fine-tuning      | **Two-phase**: (1) rote memorization in unsupervised training, (2) generalization with supervised training | Fully synthetic fact triplets; synthetic meaningless token in phase 1 and meaningful prompts in phase 2    | **LLMs can reinterpret memorized facts and generalize from memorized data**; deeper memorization improves generalization; phase (2) generalization leads to changes in embedding structure. |

---

> > > > ### Author Response · Authors · 2025-11-26
> > > > **Continue learning baseline reduces forgetting, but it learns new knowledge substantially less effectively and requires significantly more resource**
> > > >
> > > > # For the continual learning baseline:
> > > >
> > > >
> > > > We thank the reviewer for the clarification. We did not intend to imply that continual learning as a field aims to “avoid memorization.” Our wording referred specifically to certain continue learning approaches that reduce overfitting to newly encountered data, such as those that replay previously observed samples during training.
> > > >
> > > > **Our memorize-then-generalize framework is motivated by a different goal**: understanding how large language models (LLMs) acquire new factual knowledge by leveraging previously memorized information to enable generalization. Analyses of old-knowledge performance (i.e., forgetting) are therefore not the focus of our work. Instead, they serve to illuminate the learning dynamics induced by our training procedure. This analysis complements, but is not central to, our main claim that LLMs can generalize from memorized data to new facts.
> > > >
> > > > That said, we agree that comparisons to continue learning methods incorporating replay can strengthen the discussion. Accordingly, we added a preliminary experiment comparing our framework with a supervised fine-tuning (SFT)–based continual learning baseline. In this baseline, continual learning is implemented solely by replaying old facts during SFT, that is, the model is fine-tuned jointly on the new facts and on previously observed real-world facts. The results, summarized below for both generation and multiple-choice accuracy, show that while SFT with replay reduces forgetting, it learns new knowledge substantially less effectively and requires significantly more resources (more prompts and more computation).
> > > > ## Baseline setup:
> > > > ### Models and Data
> > > > We conduct experiments using the Qwen2.5-1.5B base model. Across all settings, we use five relations, each associated with 100 injected new facts. Old knowledge corresponds to the model’s pretrained real-world facts for the same relations.
> > > >
> > > >
> > > > ### Continual Learning Baseline: SFT with Replay
> > > > For comparison, we include a continual learning baseline implemented via supervised fine-tuning (SFT) with replay:
> > > >
> > > >
> > > > - The model is fine-tuned jointly on 100 new facts and 100 old real-world facts per relation.
> > > > - Training uses 10 prompts per relation.
> > > > - This baseline corresponds to a standard replay-based continual learning strategy.
> > > >
> > > >
> > > > ### Training Details
> > > > - Two learning rates are considered: 3e-06 and 1e-06.
> > > > - All experiments use the same five relations.
> > > >
> > > >
> > > > ### Evaluation Metrics
> > > > We evaluate both the generation accuracy and multiple choice accuracy as we described in the paper line 210-215.
> > > > Each result is reported as mean ± standard deviation across different prompts and relations.
> > > >
> > > >
> > > > Measured by generation accuracy:
> > > >
> > > > Base model (Qwen2.5-1.5B) for old knowledge: 0.10 ± 0.02
> > > > | Method                          | Learning Rate | New Knowledge       | Old Knowledge       |
> > > > |---------------------------------|---------------|----------------------|----------------------|
> > > > | **Our framework**| 3e-06 | 0.85 ± 0.02 | 0.10 ± 0.02 |
> > > > |                                 | 1e-06        | 0.07 ± 0.01          | 0.17 ± 0.03          |
> > > > | **SFT with replay** | 3e-06 | 0.19 ± 0.08 | 0.19 ± 0.13 |
> > > > |                                 | 1e-06        | 0.04 ± 0.02          | 0.21 ± 0.12          |
> > > >
> > > >
> > > >
> > > >
> > > > Measured by multiple-choice accuracy:
> > > >
> > > > Base model (Qwen2.5-1.5B) for old knowledge: 0.58 ± 0.06
> > > > | Method                          | Learning Rate | New Knowledge       | Old Knowledge       |
> > > > |---------------------------------|---------------|----------------------|----------------------|
> > > > | **Our framework** | 3e-06 | 0.91 ± 0.02 | 0.42 ± 0.07 |
> > > > |                                 | 1e-06        | 0.23 ± 0.02          | 0.55 ± 0.06          |
> > > > | **SFT with replay** | 3e-06 | 0.33 ± 0.10 | 0.57 ± 0.25 |
> > > > |                                 | 1e-06        | 0.16 ± 0.05          | 0.59 ± 0.25          |

---

### Official Review · Reviewer_D3F1 · 2025-11-01

**Soundness:** 3
**Presentation:** 3
**Contribution:** 3
**Rating:** 6
**Confidence:** 4

**Summary:**

This paper challenges the conventional view that rote memorization (verbatim learning through repetition) is harmful to an LLM's ability to generalize. The authors argue that LLMs can, in fact, generalize effectively from data they have rote-memorized. They introduce a two-phase "memorize-then-generalize" framework to demonstrate this. The key finding is that the model learns to reinterpret the [X] token as meaning "mother of" and applies this semantic understanding to all other associations it memorized in Phase 1. This generalization extends to unseen prompts, different facts, and even other languages. The analysis shows that during Phase 2, the model's internal representation of the key token aligns with the representations of the meaningful prompts.

**Strengths:**

1. The paper presents a more efficient and effective method for knowledge injection than standard SFT or ICL .
2. The decision to use a fully synthetic dataset with fictional entities is a major strength. This eliminates the confounding variable of pre-existing knowledge and ensures that the model is learning the facts entirely from the training, making the findings on knowledge acquisition highly reliable.
3. The findings are shown to be consistent across 8 different LLMs from 4 model families, including both base and instruction-tuned models. This strongly suggests the "memorize-then-generalize" phenomenon is a fundamental property of these architectures, not an artifact of a single model.

**Weaknesses:**

1. A primary goal of knowledge injection is to update or correct existing, incorrect facts stored in a model's parameters. The paper's framework is not tested in this more realistic and challenging scenario. It's unclear if the "memorize-then-generalize" method would be effective, or perhaps even detrimental, when the rote-learned fact (e.g., Paris [X] Germany) conflicts with strong pre-trained knowledge.

**Questions:**

1. How does the "memorize-then-generalize" framework perform when the new fact (e.g., Subject [Y] New_Object) conflicts with a fact the model already "knows" (e.g., Subject [pre-trained relation] Old_Object)? Does the Phase 1 rote memorization of the new association successfully override the old one, or does the pre-trained knowledge interfere?

---

> ### Author Response · Authors · 2025-11-20
> **Training on conflicting facts do not lead to clean knowledge replacement but instead produce unstable behavior.**
>
> We thank the reviewer for the thoughtful feedback. We are encouraged that the reviewer finds the paper well-executed and the findings interesting.
>
> Regarding the concern about knowledge updating, we would like to clarify that **our framework is not designed to update or correct existing knowledge**. Instead, our focus is on **the investigation of the phenomenon that newly introduced information can generalize after a controlled memorization phase**. For this reason, we intentionally avoid scenarios in which the injected facts directly conflict with strong pre-trained knowledge.
>
> While this setting lies outside the core goals of our framework, we agree that examining such cases offers a useful perspective. To address this, we ran preliminary experiments in which we intentionally injected conflicting facts (e.g., ``Paris is in Germany'') during the memorization phase and then applied the generalization phase. We have added these experiment details and results to Appendix G.4 in the revised manuscript.
>
> ## The results show that training on conflicting facts does not lead to clean knowledge replacement but instead produces unstable behavior.
>
> We selected five relations from the T-REx [1] dataset—*instance of, mother, capital, educated at, and author*—and sampled 400 factual triples for each. For every triple, we constructed a conflicting counterpart by replacing the original object with an incorrect one (e.g., modifying “Germany–capital–Berlin” to “Germany–capital–Paris”). We then trained the Qwen2.5-1.5B model using the memorization-then-generalization framework on this conflicted dataset. After training, we evaluated the final model on all test prompts (10 per relation), measuring accuracy with respect to both the original correct facts and the altered, incorrect variants.
>
> For the base model Qwen2.5-1.5B, only 30.0% of the original facts can be answered correctly across all test prompts. We therefore divide the original dataset into known (memorized) and unknown (non-memorized) subsets based on this.
>
> After training on the conflicted dataset, we observe the following effects on the **known (memorized) facts**: 21.4% of them are not forgotten but produce neither the correct nor the conflicted answer; 29.1% generate only the conflicted answer; 37.2% produce both the correct and conflicted answers; and only 12.2% retain the correct answer exclusively, without being overwritten.
>
> For the **unknown (non-memorized) facts**, 55.6% produce the conflicted object after training, while 37.2% fail to generate either the correct or conflicted answer. Interestingly, 7.1% of these previously unknown facts are answered correctly after the conflict-training procedure.
>
> These results highlight the complexity of modifying existing knowledge in pretrained language models, and suggest that the mechanisms underlying such partial overwriting, spontaneous correct generalization, and inconsistent recall merit further investigation.
>
> [1] Hady Elsahar, Pavlos Vougiouklis, Arslen Remaci, Christophe Gravier, Jonathon Hare, Frederique Laforest, and Elena Simperl. T-rex: A large scale alignment of natural language with knowledge base triples. In Proceedings of the Eleventh International Conference on Language Resources and Evaluation (LREC 2018), 2018.

---

### Meta-Review · Area_Chair_2NvT · 2026-01-11

**Summary:**

This paper was reviewed by four experts in the field. The recommendations are (2, 6, 6, 8). Most of the reviewers appreciate the contribution of the proposed solution and recommend acceptance of the manuscript. Based on the reviewers' feedback, the decision is to recommend the acceptance of the paper. The reviewers did raise some valuable concerns (especially more detailed experimental comparisons and ablation studies raised by all four reviewers, clearer paper presentation, and more descriptions regarding methodological insight and motivation raised by Reviewer BDrL and 8CfY) that should be addressed in the final revised version of the paper. The authors are encouraged to make the necessary changes to the best of their ability.

**Reviewer Concerns:**

During the rebuttal phase, the authors successfully addressed primary concerns related to design motivation, presentation quality, and experimental validation. However, the depth of the analysis and discussion remains an area for improvement. We strongly recommend further elaboration in these sections to fully substantiate the paper's arguments and ensure the final revision is robust and convincing.

**Reviewer Scores:**

While the reviewers may appreciate the improvements in presentation, motivation, and comparative experiments, they may maintain that the analytical depth remains insufficient. To fully validate the findings and address the complexities of the method, the discussion section must be expanded in the final revision.

---

### Decision · Program_Chairs · 2026-01-26

Accept (Poster)